**METHOD**

# Leveraging neighborhood representations of single-cell data to achieve sensitive DE testing with miloDE

Alsu Missarova[1,2], Emma Dann[3], Leah Rosen[1,2], Rahul Satija[4,5*] and John Marioni[1,2*]

*Correspondence:
rsatija@nygenome.org;
marioni@ebi.ac.uk

[1] European Molecular
Biology Laboratory, European
Bioinformatics Institute,
Wellcome Genome Campus,
Cambridge, UK
[2] Cancer Research UK
Cambridge Institute, University
of Cambridge, Cambridge, UK
[3] Wellcome Sanger Institute,
Wellcome Genome Campus,
Hinxton, Cambridge, UK
[4] Center for Genomics
and Systems Biology, NYU, New
York, USA
[5] New York Genome Center, New
York, USA

## Abstract

Single-cell RNA-sequencing enables testing for differential expression (DE) between conditions at a cell type level. While powerful, one of the limitations of such approaches is that the sensitivity of DE testing is dictated by the sensitivity of clustering, which is often suboptimal. To overcome this, we present miloDE—a cluster-free framework for DE testing (available as an open-source R package). We illustrate the performance of miloDE on both simulated and real data. Using miloDE, we identify a transient hemogenic endothelia-like state in mouse embryos lacking Tal1 and detect distinct programs during macrophage activation in idiopathic pulmonary fibrosis.

## Background

Diseases are complex and dynamic processes, with different organs, tissues, and cell types undergoing unique changes that often manifest at the transcriptional level. Single-cell genomics provides a sensitive and unbiased lens for identifying how cell type-specific phenotypes change following a perturbation or in disease [1–3]. To perform a comparative analysis between conditions, it is necessary to employ a case–control experimental design that requires the existence of several replicates from the control (typically healthy or wild type samples) condition together with replicates with the case phenotype (typically disease or perturbation). The standard workflow to identify potential case-specific phenotypes involves the batch-corrected embedding of all samples on the same low-dimensional space followed by comparative analysis between "comparable," transcriptionally similar cells (e.g., within the same cell type) [4–6]. Examples of such comparative analyses include differential abundance (DA) testing—a framework to estimate whether different cell types change their relative abundance between conditions, and differential expression (DE) analysis, in which individual genes that are expressed at different levels between conditions are identified.

DE analysis, at both the bulk and single-cell level, has yielded valuable insights into the mechanisms of numerous diseases [7–9] and enabled the identification of drug targets [10, 11]. Accordingly, numerous methods for differential expression testing have been proposed and benchmarked [12–16]. Methods originally developed for bulk RNA-seq, such as limma-voom [17], edgeR [18–20], and DESeq2 [21], were adapted for scRNA-seq analysis, by aggregating mRNA counts from transcriptionally similar cells, yielding so-called "pseudo-bulk" counts. As scRNA-seq has become increasingly popular, methods designed to handle specific properties of scRNA-seq data have been proposed (SCDE, scDD, D3E, MAST, DECENT, etc.) [22–26]. The main factor distinguishing these methods is the assumed underlying distribution of the expression counts (the main being the Poisson and the Negative Binomial, as well as the zero-inflated versions of those). Finally, a recently proposed approach, lvm-DE, uses a Bayesian framework leveraging posterior distributions estimated from deep generative models, which is suggested to be suitable for the complex, non-linear experimental designs that are particularly relevant for performing DE analysis between groups in extensive cohort studies with complex metadata [27].

Importantly, all of these methods require that cells are grouped into presumably homogenous, transcriptionally similar clusters (i.e., cell types). This minimizes variability in gene expression counts between cells within a sample, mitigates the inflation of *p*-values, and increases the power to detect differential expression for lowly expressed genes [28]. However, performing statistical tests at the cell type level (i.e., asking whether a gene is differentially expressed between conditions within a specific cell type) is ultimately limited and dictated by the sensitivity and resolution of the cell type annotation, which is a highly subjective and study-dependent process. As a result, genes that are DE only within a certain homogenous sub-region within the cell type (with the cell type either consisting of several discrete subpopulations, continuous trajectories, or a combination of both), will often be undetected. On the other hand, if sub-cell type composition differs between case and control samples, genes that are specific in their expression to a local sub-region of a cell type (in both conditions) might get falsely identified as showing significant DE when, in fact, they "mark" DA sub-regions. Finally, due to potentially substantial differences in cell type abundances, the power to identify statistically significant differences in expression can vary across cell types.

These shortcomings motivate the development of frameworks that are more sensitive to the local transcriptional structure, and that will learn the per gene DE on the manifold rather than separately for each annotated cell type. For example, the recently proposed computational suite Cacoa incorporates a DE framework that is performed at a single-cell level [29]. However, it does not account for potential confounding covariates and batch effects, making it challenging to apply in many settings, including cohort studies and complex experimental designs which are becoming increasingly popular.

To overcome these limitations, we present miloDE—a cluster-free framework for DE testing. We extend Milo, a method for cluster-free DA testing, where cell abundance is estimated over overlapping neighborhoods on the k-Nearest Neighbors (henceforth kNN) graph representation of scRNA-seq data [30]. We address key differences between DA and DE testing at both the neighborhood assignment level and when performing the statistical testing and multiple hypothesis testing correction, allowing us to perform DE

detection for each gene and each neighborhood. Importantly, the fine neighborhood resolution of miloDE unlocks a suite of methods tailored for scRNA-seq analysis, enabling the detection of co-regulated transcriptional modules containing genes that change their expression in a coordinated manner. Finally, we demonstrate the performance of miloDE in both simulated data and in different biological contexts. miloDE is an open-source R package, available at https://github.com/MarioniLab/miloDE.

## Results

### Overview of the method

miloDE is a cluster-free framework for differential expression (DE) testing that leverages a graph representation of scRNA-seq data (Fig. 1). To construct a graph recapitulating distances between cells, we require count matrices and a pre-calculated joint latent embedding across all the replicates from tested conditions (Fig. 1, step 1). Following the graph construction, we assign cells in the neighborhoods, by randomly selecting a subset of cells as neighborhood centers (referred to as index cells), subsequently assigning each index cell along with its neighbors to a single neighborhood (Fig. 1, steps 2–3; Methods). On a biological level, a single neighborhood represents a small group of highly similar transcriptional cells (i.e., one transcriptional state). Therefore, a neighborhood

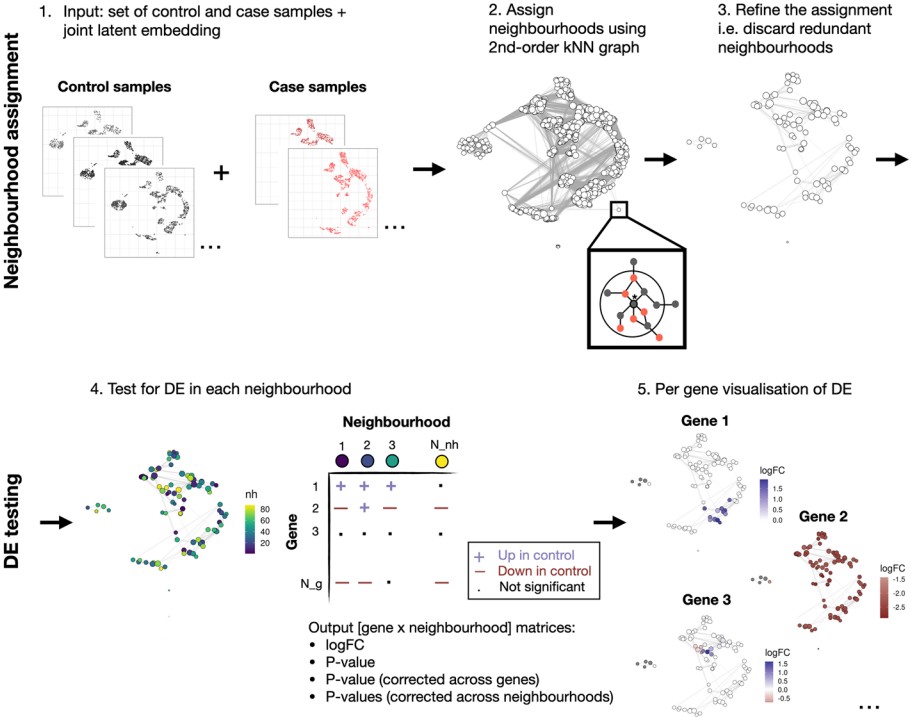

**Fig. 1** Schematic overview of the method. As input the algorithm takes a set of samples with given labels (case or control) alongside a joint latent embedding (step 1). Next, we generate a graph recapitulating the distances between cells and define neighborhoods (step 2) using the 2nd-order kNN graph (inset). We then refine the neighborhood assignment and discard redundant neighborhoods (step 3). In step 4, we index our neighborhoods, and then proceed with DE testing within each neighborhood. As an output, we return four [gene × neighborhood] matrices, corresponding to logFC and statistics, raw and corrected either across genes or across neighborhoods. Results for each gene can be visualized using neighborhood plots (step 5), in which each neighborhood (circle) is colored with the estimated logFC if significant

assignment provides a representation of single-cell data as a set of local transcriptional states. We note that neighborhood representation is different from metacell representations such as MetaCell [31] or SEAcells [32] since it overcomes the limitation of discretization and allow neighborhoods to overlap (i.e., each cell can be assigned to multiple neighborhoods). This provides a more continuous and connected representation of transcriptional states.

Once neighborhoods are assigned, we test for DE within each neighborhood (Fig. 1, steps 4–5). To do so, for each neighborhood, we sum the counts for each replicate and use the edgeR framework—a scalable and highly performant DE detection method that utilizes a generalized linear model framework that allows the incorporation of covariates and complex experimental designs [18, 33]. As an output, we return [gene × neighborhood] matrices containing the estimated log2 Fold Change (logFC) and statistics indicating the significance of each comparison within each neighborhood. We then perform multiple testing corrections in two directions: for each neighborhood, we correct across all tested genes, and for each gene we correct across all tested neighborhoods, accounting for neighborhood sizes (i.e., the number of cells per neighborhood) and the overlap between neighborhoods (see Methods). This dual correction scheme allows the identification of both DE-neighborhoods for specific genes and DE genes for specific neighborhoods.

Having outlined the method, it is essential to recognize that DE detection is dependent on how we group cells prior to testing. Below we discuss in greater detail some key aspects of the neighborhood assignment (steps 1–3) and how they potentially affect DE detection.

### Choice of the latent embedding

As mentioned above, we require a pre-calculated latent embedding as an input. Although the choice of the embedding/integration technique is left to the user, we note that "supervised" and "unsupervised" approaches result in neighborhood assignments that impact DE detection in different ways. We suggest that in the "supervised" settings (e.g., Azimuth [4] and scArches [5]), in which underlying variance is learned solely from the control samples followed by the transfer of the case samples using the learned model, the input from case-specific variance to the embedding is minimized. In contrast, in the "unsupervised" settings, where variance is jointly learned from control and case samples (e.g., mutual nearest neighbors (MNN) [34] and scVI [35]), the case-specific variance may contribute to the embedding. The potential consequence of this is that cells will separate on the embedding by the aforementioned case-specific variance thus hindering our ability to aggregate them in the same neighborhoods and detect DE for this specific variance. Combined, we hypothesize that "supervised" integration approaches are more suitable for sensitive DE detection. Moreover, we show via simulations that DE detection in the "supervised" embeddings is quantifiably more sensitive (see in-depth discussion and supporting analysis in Additional file 1: Supplementary Note 1 [36–38]).

### Graph construction

By design, miloDE aims to detect DE on a local level, within individual transcriptional states. Having more cells within the neighborhoods increases the probability of

generating more heterogeneous neighborhoods, thus hindering the ability to capture distinct transcriptional states. Accordingly, ideal neighborhood assignments should contain neighborhoods of the smallest possible size. On the other hand, the power to detect DE using pseudo-bulk approaches such as edgeR is highly dependent on and scales with the number of tested cells [14]. We explore the importance of a sufficient number of cells as well as other parameters such as the number of tested replicates on DE detection in Additional file 1: Supplementary Note 2 (summarized in Additional file 2: Table S1). Importantly, we estimate that if the total number of tested cells is around 350, the sensitivity to detect logFC = 1 is ~ 65% (~ 85% for logFC = 2). Therefore, if we aim to achieve sensitivity above 65%, ideally we would require our neighborhoods to contain several hundred cells on average. A standard approach for constructing a graph representation is to use a kNN graph, in which each cell is connected with its *k* nearest neighbors in the latent space. Since the number of cells per neighborhood scales nearly linearly with *k*, to achieve several hundred cells per neighborhood, we would require *k* to be in the same range. However, since kNN graph has limited sensitivity to the local density, having high values of *k* is likely to compromise the homogeneity of the neighborhoods. In other words, if *k* is high enough, cells from rare cell types (i.e., less abundant than the chosen *k*) are prone to get connected (and therefore assigned to the same neighborhoods) with transcriptionally similar but more abundant cell types. Accordingly, if a certain transcriptional change is restrained to such rare cell types, we hamper sensitive DE detection in the neighborhoods containing the mixture of the cell types. Previously, several methods have been developed to improve kNN graph representation to better recapitulate local density [31, 39, 40]. These methods aim to identify phenotypically distinct, non-overlapping, fine-grained cell states. Motivated by this, we introduce a 2nd-order kNN graph representation in which a standard (hereafter referred to as a 1st-order) kNN graph is amended with additional edges between any 2 cells that share at least one neighboring cell (Fig. 1, Step 2, inset). We suggest that the 2nd-order kNN representation returns more homogenous neighborhoods than the 1st-order kNN, while controlling for the average neighborhood size. Specifically, as *k* increases, the average neighborhood size for the 2nd-order kNN graph increases considerably faster than it does for the 1st order (Additional file 3: Fig. S1). Importantly, as *k* reaches 20–25 in the 2nd-order setting, the method will result in sufficiently large average neighborhood size (higher than 350, Additional file 3: Fig. S1). Importantly, we assume that, specifically for rare cell types, the corresponding neighborhoods will be smaller in size but considerably more homogeneous, when compared to the 1st-order kNN graph, and this is potentially more important for DE testing than neighborhood size.

To quantitatively assess the detection differences between the assignments using either 1st- or 2nd-order kNN graphs, we used a chimeric mouse embryo dataset, in which tdTomato + mouse embryonic stems cells containing a Tal1 knock out were injected into wild type blastocysts [41]. We used an atlas of wild type (WT; i.e., non-chimeric) gastrulating mouse embryos as a "control" (henceforth referred to as WT), and the wild type cells from the chimeric embryos were used as the "case" (henceforth referred to as ChimeraWT). We assigned neighborhoods using different *order* and *k* combinations while restricting *order-k* values such that the average neighborhood size was within the low hundreds cell target (Additional file 3: Fig. S2A). Additionally, a certain degree of

stochasticity, owing to the random selection of index cells, invariably impacts into any neighborhood assignment, potentially propagating into discrepancies in the DE detection. To address this, in the following simulations, for each [*order-k*] combination (hereafter referred to as an assignment) we repeated the assignment 10 times.

First, we compared the homogeneity of the neighborhoods between 1st- and 2nd-order assignments. To do so, we derived two complimentary metrics that we refer to as relative cell type enrichment and cell type purity. To calculate the cell type enrichment score, we first annotated each neighborhood with its most enriched cell type before calculating the fraction of cells from the neighborhood that is annotated with the assigned enriched cell type. We suggest that higher cell type enrichment distribution across all cell types reflects "more homogenous" assignments that more effectively segregate cell types between each other. Additionally, it provides a per cell type estimate of how well it is segregated from the other cell types. The second metric—cell type purity score—is defined as the percentage of cells from the neighborhood that have the cell type label in question. We formulated the cell type purity metric in this way since we hypothesized that for a perturbed cell type, neighborhoods with higher cell type purity score (for a given cell type) will have higher sensitivity of DE detection. We also expect that a 2nd-order kNN assignment is more likely to return more neighborhoods with higher cell type purity score for all of the cell types, including rare ones. In contrast, we assumed that a 1st-order assignment will likely "merge" rare cell types with transcriptionally similar but more abundant cell types thus making it more challenging to detect DE within those rare cell types. Following the definition of the metrics, within each neighborhood assignment and each tested cell type (Additional file 3: Fig. S2B, Methods), we calculated the relative cell type enrichment score (Additional file 3: Fig. S2C) and maximum cell type purity score (Additional file 3: Fig. S2D). We consistently observe higher cell type purity scores for the 2nd-order assignments for rare cell types, across a wide range of average neighborhood sizes and replicates of the assignments. We observe no comparable difference for the more abundant cell types, regardless of how well these cell types are segregated from the other cell types (estimated by cell type enrichment score).

Finally, we examined how the identified difference in cell type purity affects the sensitivity of DE detection. To do this, we synthetically introduced "ground truth DE" by perturbing the expression of a small number of genes per cell type (see Methods). We then applied miloDE for each neighborhood assignment and asked how the DE detection (i.e., fraction of "perturbed" genes that are detected as DE) depends on cell type purity as well as the absolute number of cells from the cell type. To do so, for each neighborhood and each "perturbed" cell type, we calculated the following measurements: cell type purity, the number of cells from the cell type, and DE detection power (Additional file 3: Fig. S3A). As expected, DE detection power scales with both cell type purity and the number of cells from the cell type. Importantly, for the rarest cell type—primordial germ cells (PGCs)—we achieve considerably high detection only for highly pure neighborhoods, which are only present in the 2nd-order assignments (maximum detection = 0.78 in 2nd-order against 0.63 in 1st-order). When we calculate the maximum DE detection power for each assignment and each cell type, we observe a small but consistently higher detection power in the two most rare cell types—PGCs and Blood progenitors—for the 2nd-order assignments (Additional file 3: Fig. S3B). Interestingly, the detection power

for these cell types, on average, decreases with higher neighborhood sizes, likely because rare cell types start to get "mixed" with other cells. Overall, we conclude that 2nd-order assignments increase the power to detect DE in rare cell types while preserving the same power for the more abundant cell types, and therefore is a more suitable choice for our framework.

### Refinement of the neighborhood assignment

Scalability is an important aspect of any algorithm, and it is important to minimize the computing time for better usability. To decrease the computing time we want to minimize the number of tests (i.e., the total number of neighborhoods) while ensuring that all cells are assigned to at least one neighborhood (i.e., for each cell, there is at least one neighborhood that it belongs to). In the original Milo approach, once the graph is constructed, the number of assigned neighborhoods is controlled by the parameter *prop*— the initial proportion of cells selected as index cells (Methods). Since it is unclear how to select the lowest *prop* while ensuring "complete coverage" (i.e., maximize the probability that all cells are assigned while avoiding the computationally demanding task of constructing a neighborhood for each cell), we introduce post hoc neighborhood refinement (Fig. 1, step 3), in which we first assign a high enough number of neighborhoods (we use *prop* = 0.2, but this can be increased by the user) to maximize the probability of the complete coverage, followed by sorting the neighborhoods in decreasing order of size and iteratively discarding neighborhoods where all cells are included in previously assigned neighborhoods (Methods). To test how post hoc filtering performs compared to selecting an optimal *prop*, we used the mouse gastrulation dataset introduced previously, constructed neighborhoods with *prop* = 0.2 (i.e., used 20% of cells as index cells), and performed post hoc filtering. Based on the number of neighborhoods after refinement, we then selected a considerably lower *prop* that resulted in a comparable number of neighborhoods (without the refinement step). We observe that in the original, "unrefined" neighborhood assignments around 10% of cells are consistently unassigned to any neighborhoods, whereas all cells are assigned in the refined assignment (Additional file 3: Fig. S4).

### miloDE enables sensitive and precise DE detection in simulated data

We next assessed the performance of miloDE in synthetically perturbed data. We used the same WT-ChimeraWT dataset introduced previously and perturbed the expression of 5 genes in one condition and in one selected cell type (henceforth "perturbed cell type") (Methods, Fig. 2A). To ensure that the selected genes are not DE elsewhere on the manifold, we chose the genes from a pool of candidate genes that are not DE in any cell type. We assigned neighborhoods and performed miloDE testing using $k$ = 20, 25, 30 (2nd-order kNN), and for each $k$ generated 5 replicates, i.e., independent neighborhood assignments (Fig. 2B). Next, for each neighborhood assignment, we calculated the per neighborhood cell type purity, i.e., how many cells from a neighborhood have the perturbed cell type label. We define neighborhoods as "ground truth" DE if cell type purity for these neighborhoods exceeds a certain cell type purity threshold for the perturbed cell type. Finally, for any "perturbed" gene and any cell type purity threshold, we can calculate the detection power for this gene

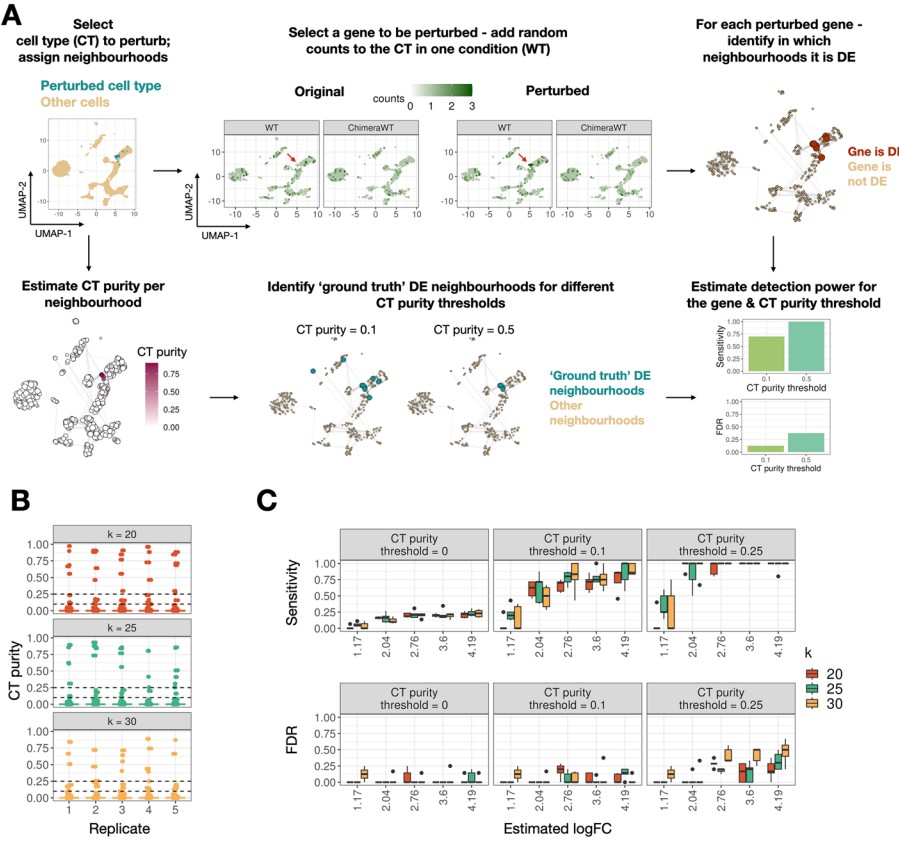

**Fig. 2** miloDE yields sensitive and precise detection in simulated data. **A** Schematic representing how detection power is estimated for each gene and cell type purity threshold. The top left panel illustrates single-cell data, embedded in UMAP space, with highlighted cell type, in which we will alter the counts. The top middle panel represents the "in silico" perturbation we introduce to the selected cells, and the top right panel represents neighborhood assignment, followed by per neighborhood quantification of whether the selected gene is identified as DE. The bottom left panel represents per neighborhood quantification of the cell type "purity." The bottom middle panels represent how "ground truth" DE neighborhoods are selected based on cell type purity threshold. The bottom right panel illustrates the final quantification of DE detection (i.e., sensitivity + FDR). **B** Boxplots representing the distribution of cell type purity score across neighborhoods for each neighborhood assignment replicate and for $k = 20$, 25, and 30. Dashed lines correspond to selected cell type purity thresholds. **C** Boxplots representing how sensitivity and FDR change with estimated logFC, $k$, and cell type purity threshold. Each box represents data across 5 replicates for a single $k$

(i.e., sensitivity and false discovery rate (FDR)) based on the overlap between "ground truth" DE neighborhoods and neighborhoods in which the gene is identified as DE. We then assessed how sensitivity and FDR change as a function of the cell type purity threshold (Fig. 2C). Consistent with the expectation that the probability of detecting DE in a neighborhood scales with both the fraction of "altered" cells as well as the effect size, we observe that sensitivity and FDR increase as a function of both cell type purity threshold and estimated logFC. Specifically, when using a cell type purity threshold of 0.1 (i.e., all neighborhoods in which at least 10% of cells belong to the perturbed cell type are flagged as "ground truth" DE), we observe average sensitivity 0.73 with logFC > 2. Therefore, we suggest that even a small fraction of perturbed cells in one condition is sufficient for a test result to be flagged as significant. Importantly, we report high sensitivity with well-controlled FDR (average FDR 0.06 with logFC > 2,

for the cell type purity threshold 0.1). Moreover, these trends are robust across different levels of *k* and independent neighborhood assignments.

We next sought to compare the performance of miloDE to another cluster-free DE method Cacoa [29]. To do so, we generated several simulations, by varying the fraction of the perturbed cell group (imitating the cases in which only a subset of cells or a sub-cell type exhibits DE) and the number of case and control replicates (Additional file 3: Fig. S5A,B, Methods). For miloDE, we performed several neighborhood assignments ($k = 20, 25, 30$; 5 independent assignments for each level of *k*). For each simulation, we selected several genes and added counts in silico to the case samples to create a set of "ground truth" DE genes with a wide range of the estimated effect size. In Cacoa, DE is estimated for every cell, and for each cell, the estimated logFC is returned alongside the *z*-score statistic (both raw and adjusted for the multiple testing correction).

Since the outputs of miloDE and Cacoa are not directly comparable (miloDE returns statistics per neighborhood while Cacoa returns statistics per single cell), we approached the comparison in two ways. First, we aggregated the *z*-scores across the neighborhoods assigned with miloDE (using the neighborhood assignments with $k = 25$, for each neighborhood—average across all the cells from the neighborhood). In the absence of "ground truth" DE for the neighborhoods, we used two thresholds (0.1 and 0.25) for the perturbed group fraction purity (akin to cell type purity) to decide which neighborhoods contain enough cells to be considered "ground truth" DE. This allowed us to estimate the area under the curve (AUC) for each approach, perturbed gene, and the designated perturbed group purity threshold (we aggregated AUCs across different neighborhood assignments for the same *k*, Additional file 3: Fig. S6A). Overall, for miloDE, we observe high AUCs across all tested conditions (average AUC 0.95 for the purity threshold 0.1 and 0.99 for the purity threshold 0.25), except for a dataset with only 4 replicates in total and an extremely small percentage of cells (0.5%) being perturbed. Additionally, given the stochasticity in the neighborhood assignments due to the random selection of index cells, we assessed the robustness of the performance for different neighborhood assignments for the same *k* (Additional file 3: Fig. S7). We observe highly similar performance across most of the conditions, with the exception of genes with low effect size in datasets with a very low number (below 5) of "ground truth" DE neighborhoods (Additional file 3: Fig. S7A). We suggest that in these settings, even a small change in the number of "ground truth" DE neighborhoods is expected to greatly skew the sensitivity of the detection.

As for Cacoa, we observe a striking difference when we compare the performance using raw or adjusted *z*-scores. Cacoa returns extremely high AUCs (higher than miloDE) while using raw *z*-scores, reflecting the fact that the aggregation of *z*-scores across cells within a neighborhood is more sensitive to the fraction of perturbed cells in a neighborhood. However, the multiple testing correction in Cacoa appears to be highly sensitive to the overall fraction of perturbed cells, highlighting the challenging task of the correction across a large pool of tests (i.e., across every cell). We observe that for datasets with a perturbed fraction below 5%, adjusting *z*-scores in Cacoa eliminates any significance (AUC = 0.5), while AUCs in miloDE remain high even for the datasets with the smallest perturbed fraction.

We also aimed to compare miloDE and Cacoa on a single-cell level. For this, we "decomposed" the output of miloDE to return the statistic for each cell. To do so, we estimated a *z*-score for each cell from miloDE output, by first transforming corrected across neighborhoods *p*-values into *z*-scores, and then, for each cell, taking the average *z*-score across all neighborhoods to which it was assigned (Additional file 3: Fig. S6B). In this comparison, cells with ground truth DE are directly known from the simulations. By design, Cacoa returns a *z*-score equal to 0 for cell/gene combinations with no expression, and so to perform a fairer comparison, we further estimated Cacoa's performance using only case perturbed cells as ground truth DE or using all cells from the perturbed group as ground truth DE. We note that, as above, we observe the same significant drop in the performance of Cacoa while using adjusted *z*-scores. As for the raw *z*-scores, using only case perturbed cells as ground truth DE returns higher AUCs (average AUC 0.87 for case cells against average AUC 0.72 for all cells); however, they are still lower than the AUCs estimated for "decomposed" output from miloDE (average AUC 0.93). We suggest that this can be potentially explained by the sensitivity of Cacoa to the inherited noise in single-cell data. In contrast, miloDE allows to estimate "local" DE while simultaneously controlling for such single-cell noise, thus providing a certain rigor in the analysis.

Finally, we compared the performance of miloDE to the pseudo-bulk approach (Additional file 3: Fig. S8, Methods). As expected, when the fraction of perturbed cells is small (below 5%), even a high effect size cannot be detected by the pseudo-bulk approach (Additional file 3: Fig. S8A). Moreover, even when a perturbed gene is detected as significant for the datasets with a higher fraction of the perturbed cells, the reported effect size is considerably smaller than the designed effect size for the perturbed group (Additional file 3: Fig. S8B). Such genes might rarely get prioritized in real-life analyses since it is common to discard genes with the low effect sizes. On the other hand, while we estimate the per-gene sensitivity of miloDE detection (across neighborhoods), we observe that we can successfully detect most genes as DE in the neighborhoods in which we expect them to be DE (i.e., high sensitivity, based on the "perturbed group purity" threshold). We also observe that the reported effect sizes match considerably better the expected effect size than the pseudo-bulk approach.

Overall, we conclude that miloDE provides robust, sensitive, and precise DE detection across a wide range of effect sizes.

### Identification of hemogenic endothelial-like cells undergoing ectopic cardiomyogenesis in the absence of TAL1

Having established the good performance of miloDE, we applied it in the context of continuous developmental trajectories, where discrete clustering and cell type annotation is suboptimal. Tal1 (SCL) is a DNA-binding transcription factor that plays a key role in hematopoiesis, with mouse embryos that carry a double knockout of *Tal1* dying around embryonic day E9.5 from severe anemia [42]. Previously, the molecular function of Tal1 has been investigated using chimeric mouse embryos where *Tal1* − / − mouse embryonic stem cells are injected into wild-type blastocysts; the wild type blastomeres are able to generate blood cells, meaning that the cell-intrinsic impact of Tal1 can be effectively studied [41, 43]. As shown previously, in *Tal1* − chimeras the *Tal1* mutant cells are depleted of erythroid cells (Fig. 3A, Additional file 3: Fig. S9A). Additionally, a laborious

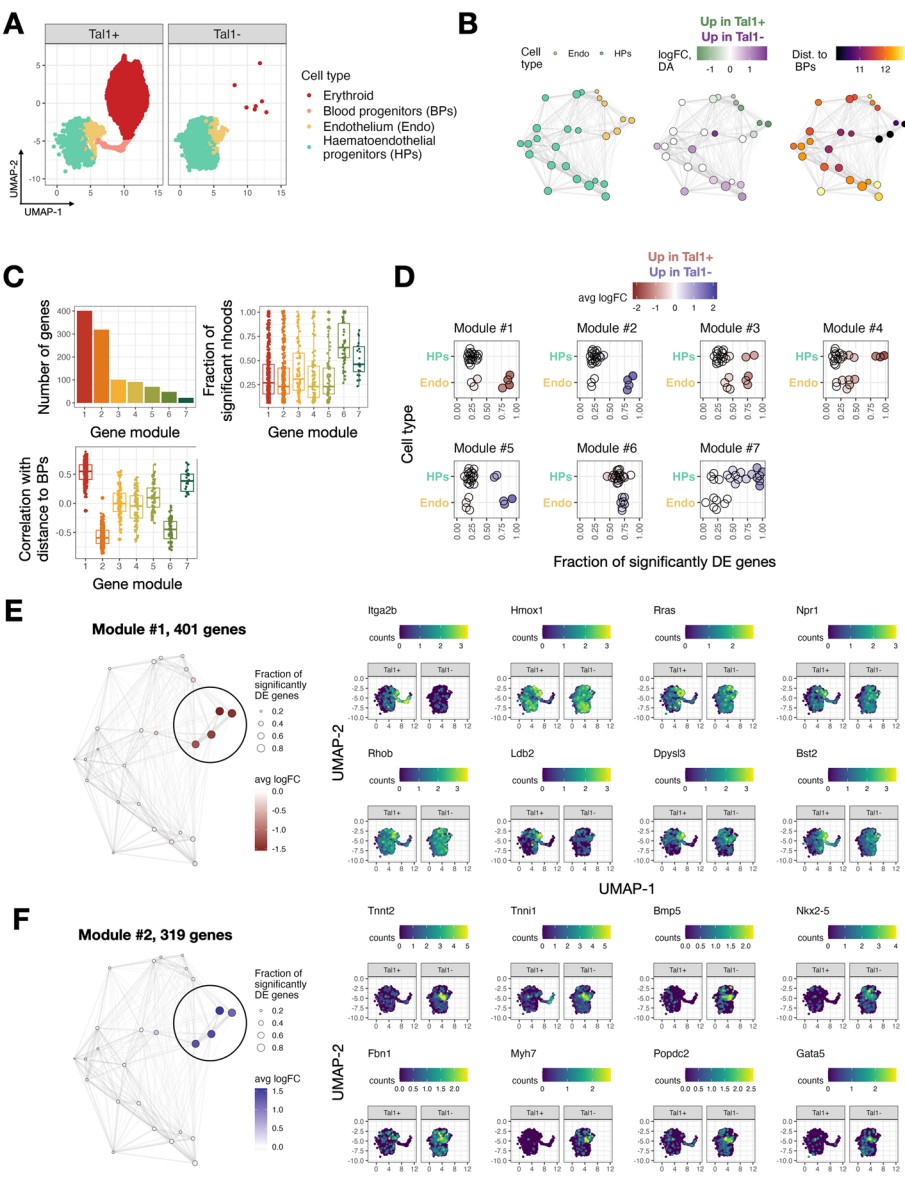

**Fig. 3** miloDE is suitable for continuous trajectories and recovers a transient transcriptional state of hemogenic endothelia. **A** UMAP representing a manifold of chimera mouse embryos, colors correspond to the cell types contributing to a blood lineage, and facets correspond to whether cells carry a knockout of Tal1. **B** Neighborhood graphs covering hematoendothelial progenitors and endothelial cells, colored by enriched cell type (left), differential abundance (middle) and distance to blood progenitor cells in PC space (right). **C** Top left panel: Barplot representing how many genes are associated with each module. Top right panel: Distribution of fraction of significantly DE neighborhoods for each module. Bottom panel: Distribution of correlation between logFC and distance to blood progenitors for each module. **D** Jitter plot representing the relationship between gene modules and cell types. Each facet corresponds to one module, each point corresponds to one neighborhood. *X*-axis corresponds to a fraction of genes from the gene module that are significantly DE in the corresponding neighborhood, color corresponds to the average logFC. **E** Left: Neighborhood plot for the 1st gene module, size of the nodes correspond to the fraction of genes from the gene module that are significantly DE in the corresponding neighborhood, color corresponds to average logFC. Right: UMAPs representing hematoendothelial progenitors and endothelial cells, colored by representative genes from the 1st module that are associated with angiogenesis and lipoxygenase activity. **F** Left: Neighborhood plot for the 2nd gene module, size of the nodes correspond to the fraction of genes from the gene module that are significantly DE in the corresponding neighborhood, color corresponds to average logFC. Right: UMAPs representing hematoendothelial progenitors and endothelial cells, colored by representative genes from the 2nd module that are associated with cardiomyogenesis

annotation of endothelial cells revealed that mutant cells that transcriptionally resemble hemogenic endothelia (which in the wild-type environment would have contributed to the second wave of hematopoiesis) do not express hemogenic markers and instead show signs of cardiomyogenesis.

To investigate if miloDE could provide further insight into this biological process, we first applied it to the whole manifold (i.e., all cell types) to identify which cell types show signs of extensive transcriptional changes upon the perturbation. To rank cell types by "degree of perturbation," for each neighborhood, we calculated the number of DE genes as well as the number of DE genes that are highly specific to this neighborhood (see Methods), followed by grouping neighborhoods by the cell type they primarily contained (Additional file 3: Fig. S9B,C). Reassuringly, neighborhoods that are enriched for cells directly contributing to the blood lineage (i.e., endothelial and hematoendothelial progenitors) rank highest for both metrics (except for one neighborhood, containing a mixture of rare cell types that are only present in one condition, Additional file 3: Fig. S9D), with all other cell types showing a considerably lower degree of perturbation. This observation supports the expectation that while endothelia and hematoendothelial progenitors are present in both conditions in similar quantities, the absence of Tal1 still results in transcriptional changes that we can identify and characterize using miloDE.

To systematically assess how hematopoiesis is disrupted in cells contributing to the blood lineage, we next applied miloDE only to cells annotated as hemoendothelial progenitors and endothelia (Fig. 3A). To characterize DE-patterns, we calculated various statistics on the assigned neighborhoods, including cell type enrichment, differential abundance and average distance to blood progenitors (Methods, Fig. 3B). To further explore DE patterns within the selected cells, we applied the WGCNA framework on logFC vectors and retrieved transcriptional modules of co-perturbation—sets of genes that show similar magnitude of DE across the neighborhoods and are thus likely to be co-regulated ([44–46], Methods). We identified 6 co-regulated modules, containing different numbers of genes (Fig. 3C). Most modules contain genes that are differentially expressed in coherent sub-regions of the manifold (Fig. 3C, Additional file 3: Fig. S10). Additionally, for several modules, the associated regions of DE (i.e., neighborhoods in which the majority of the genes from the module are DE) contain neighborhoods associated with both endothelia and hematoendothelial progenitors (Fig. 3D). This likely reflects the limitation of using discrete clustering approaches to summarize the continuous trajectory of endothelial maturation.

Next, we focused on the first two modules, which are "complementary" to one another. Specifically, for each gene, we calculated the correlation between its logFC and distance to blood progenitors (across neighborhoods), and based on it, we suggest that module one contains genes that are downregulated (in *Tal1*−cells) and module two contains genes that are upregulated in the neighborhoods that are most proximal to blood progenitors. Importantly, their "location" in the manifold suggests that they may be enriched for cells that give rise to blood lineage (Methods, Fig. 3B–D). Consistent with this, we find that the first module contains the hematopoietic marker *Itga2b* [47, 48] as well as genes associated with angiogenesis and vasculature development (*Rras*, *Hmox1*, *Npr1*, and others; Fig. 3E, Additional file 4: Table S2) and the regulation of cell migration (*Rhob*, *Dpysl3*, *Ldb2*, *Bst2*, and others; Fig. 3E, Additional file 4: Table S2), which is

characteristic of the endothelial to hematopoietic transition [49]. The second module, containing genes that are upregulated in *Tal1*−cells, is heavily enriched for genes associated with cardiomyogenesis (*Tnnt2*, *Tnni1*, *Bmp5*, *Nkx2-5*, *Gata5*, and others; Fig. 3F, Additional file 4: Table S2). This is consistent with the observations from the original study as well as previous reports suggesting an alternative, cardiac, cell fate specification in mesodermal cells that lack *Tal1* [41, 50–52]. The functional annotation of the first two modules, together with the fact that associated neighborhoods are transcriptionally "close" to blood progenitors, strongly suggests that these neighborhoods consist of hemogenic endothelia that in normal circumstances give rise to blood cells during the second wave of hematopoiesis. In agreement with this, we observe a strong correlation between the distance to blood progenitors and enrichment and DA. Specifically, neighborhoods that are most proximal to blood progenitors neighborhoods are significantly enriched for Tal1 + cells, which in turn are "preceded" by the neighborhoods that are significantly enriched for Tal1 − cells. This pattern likely reflects the inability of Tal1 − endothelial cells to commit to a hemogenic identity, followed by the block in their development and acquisition of cardiac fate. Combined, our findings confirm the sensitivity of miloDE to identify transient cell states in continuous manifolds, within and across individual cell types.

### miloDE reconstructs a functionally meaningful timeline of macrophage activation during idiopathic pulmonary fibrosis (IPF)

To showcase the power of miloDE in the context of complex diseases and datasets, we next applied it to study transcriptional changes that arise in idiopathic pulmonary fibrosis (IPF). IPF is a severe, irreversible lung condition that starts with persistent inflammation of epithelial cells, which in turn triggers an inflammatory response resulting in constant wound healing and scarring of the epithelial tissues of the lungs (fibrosis) [53].

At the cell type level, macrophages are one of the key players in IPF. Macrophages are phenotypically plastic cells, with their transcriptional make-up being highly dependent on their environment and the external stimulant. Consistent with this, their function and role in fibrosis changes with disease progression [54–56]. At the early stages of the disease, characterized by the inflammation of epithelial cells lining the airways, "classically activated" macrophages are observed (Fig. 4A). As a direct response to the inflammation, these macrophages are activated by Interferon-gamma (IFN-γ) or lipopolysaccharides and produce pro-inflammatory cytokines. In turn, a persistent inflammatory response triggers aberrant wound healing and fibrotic remodeling, associated with "alternatively activated" macrophages that can be activated by IL-4/13.

On a molecular level, one axis of macrophage variability in IPF is associated with the expression of the fibrotic marker Spp1 [8, 57, 58]. The abundance of an Spp1-high subpopulation has been repeatedly observed in IPF patients; however, Spp1-high macrophages also exist in healthy patients (albeit at a lower abundance and absolute Spp1 expression). We propose that miloDE is a suitable tool to study complex DE patterns with respect to various axes of disease-driven variability such as DA (by paired Milo—miloDE analyses) or Spp1 expression. Additionally, while using miloDE output, we can employ clustering algorithms to characterize various "co-regulated" DE patterns arising in this population of cells during disease progression.

To this end, we analyzed an existing lung atlas available within Azimuth, considering four datasets that contain cells from healthy and IPF donors (Fig. 4B). In each of these, we only considered cells annotated as macrophages. First, we assessed DA across neighborhoods within the macrophage manifold, observing (as expected) differences in abundance that are highly correlated with average Spp1 expression (Fig. 4C), suggesting that the logFC of DA can be used as a proxy for disease progression.

We then applied miloDE and characterized different DE patterns by applying Louvain clustering to group genes upregulated in at least 25% of neighborhoods into gene sets, using their logFC values across neighborhoods as a feature vector ([59], Methods). This analysis identified 5 gene sets (Fig. 4D, Additional file 5: Table S3); interestingly, neighborhoods strongly associated with most of the gene sets were enriched for specific stages of disease progression (Fig. 4E, first panel). Specifically, all "neighborhood markers" for the IPF-1 gene set have positive logFC-DA (enriched in healthy donors), and "neighborhood markers" for the IPF-2 and IPF-3 gene sets have logFC-DA around 0 (neither enriched for IPF nor healthy donors). "Neighborhood markers" for the remaining gene sets are mainly enriched in IPF donors. To further characterize gene sets statistically with respect to DA, we assigned each gene into one of three groups: DE in neighborhoods that are enriched for IPF cells, DE in neighborhoods that have both IPF and healthy cells in comparable amounts, and DE in both groups of neighborhoods (Fig. 4E, second panel; Methods). We observe that gene sets that are associated with neighborhoods that are strongly enriched for IPF cells (i.e., IPF-4 and IPF-5) mostly contain genes that are identified as DE predominantly in IPF-enriched regions of the manifold, whereas the first three sets contain a mixture of genes from different groups.

Of note, even though we focused our analysis on genes that are upregulated in at least 25% of the neighborhoods, ~21% of the genes that we used for Louvain clustering

(See figure on next page.)

**Fig. 4** miloDE enables sufficient resolution to discover macrophage-specific gene sets, specific for different points of the IPF progression. **A** Schematic cartoon (based on a figure from Zhang et. al, 2018) representing the phenotypical heterogeneity of activated macrophages upon the response to injuries in epithelial tissues. **B** Barplot representing donor composition across different datasets and conditions. **C** Top left panel: The neighborhood graph covering macrophages in both healthy and IPF patients, each node is colored by estimated differential abundance. Bottom left panel: The neighborhood graph covering macrophages in both healthy and IPF patients, each node is colored by average Spp1 expression. Middle right panel: High negative correlation (across neighborhoods) between differential abundance and average SPP1 expression reflects enrichment of SPP1 fibrotic macrophages in IPF patients. **D** Neighborhood graphs representing transcriptional profiles for IPF-upregulated gene sets. For each gene set, the size of the nodes corresponds to the fraction of genes from the gene set that are significantly DE in the corresponding neighborhood, and color represents the average logFC. **E** First (leftmost) panel: Boxplots representing the distribution across "marker" neighborhoods for each cluster (*y*-axis) with respect to logFC of the differential abundance (*x*-axis). The asterisks signify a significant difference between groups of gene sets (*p*-value < 0.05, Wilcoxon rank test). Second panel: Barplot representing a composition of different gene categories across gene sets. Third panel: Barplots representing whether genes are detected as DE in the pseudo-bulk (across whole cell type) testing. Fourth panel: Boxplots representing the distribution of the fraction of significantly DE neighborhoods for each gene, grouped by gene set. **F** Dotplot representing, for each gene set, its aggregated DE pattern along the fibrotic phenotypical progression (fibrotic progression is estimated from logFC of differential abundance testing). Color of the dots corresponds to aggregated average logFC and the size of the dots corresponds to the average fraction of genes from the gene set that are significantly DE in the corresponding neighborhood. **G** A heatmap representing functional annotation across gene sets. *Y*-axis corresponds to enriched gene ontologies, *x*-axis corresponds to the gene sets, and color corresponds to the significance of the gene ontology (GO) enrichment for the corresponding cluster

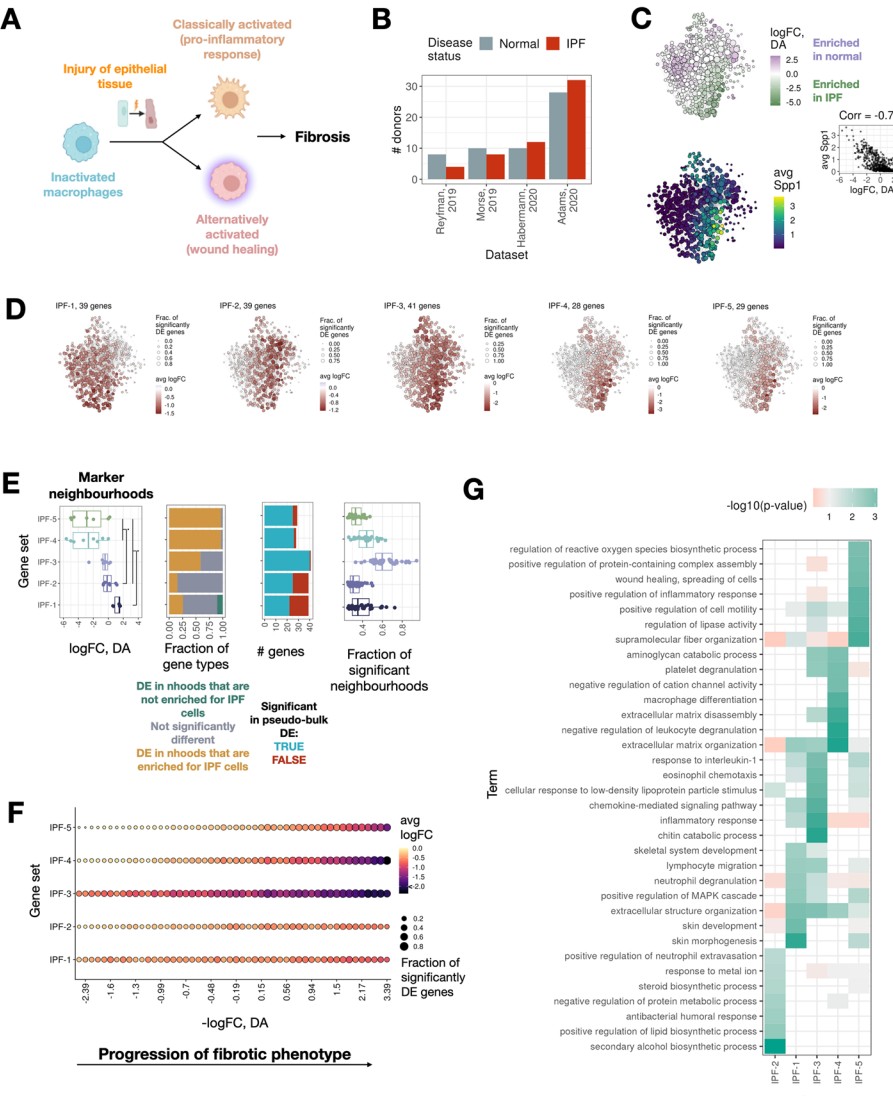

**Fig. 4** (See legend on previous page.)

were identified as not significant when DE was performed on the pseudo-bulk level (38 out of 176 genes, FDR > 0.1, across all macrophages), and this under-calling is particularly prominent in the first two gene sets (Fig. 4E, third panel; Additional file 3: Fig. S11). This "undercalling" is likely due to these genes being DE only in a subset of macrophages (Fig. 4E, fourth panel). Additionally, we systematically observe that the effect size detected by miloDE (averaged across significantly DE neighborhoods) is higher than the effect size estimated from the pseudo-bulk approach (Additional file 3: Fig. S11). Among these genes are members of the immunoglobulin family (*Iglc2, Ighg1, Ighg3, Ighg4*), genes associated with the defense response against bacteria (*Plac8, Defb1*), and chemokine response genes (*Ccl4, Ripor2*). Overall, we detect that around 1000 genes that are significantly DE in at least 5% of neighborhoods are not detected as significant while using the pseudo-bulk approach (Additional file 3: Fig. S12A, Methods). On the other hand, for 1400 genes that are identified as significantly DE with the logFC > 0.5

while using the pseudo-bulk approach, we do not detect any significant neighborhoods with miloDE. We suggest that there are several potential factors contributing to this discrepancy. On the one hand, the DE detection power of miloDE is limited by the number of cells in the neighborhoods as well as the burden of multiple testing correction across neighborhoods, and these aspects can be "partially salvaged" by aggregating more cells as inherently done within a pseudo-bulk approach. On the other hand, pseudo-bulking across many cells can be affected by the sporadic high counts in a limited and randomly distributed (across latent space) number of cells, and using miloDE will at least partially control for this (since each neighborhood is less likely to get a sufficient number of such sporadically expressed cells). To assess how likely it is that detected genes are sporadically expressed, we split genes detected as DE while using pseudo-bulk in two classes based on whether or not we detect any significantly DE neighborhoods in miloDE (corrected across neighborhoods $p$-value < 0.1), and for each gene, we calculated in how many healthy and disease cells we detect an expression. Strikingly, we observe that genes that are "not detected" in miloDE (i.e., 0 significant neighborhoods) show a considerably smaller number of expressed cells (Additional file 3: Fig. S12B). This is consistent while controlling for the logFC reported from the pseudo-bulk analysis. On the other, to estimate whether genes that are not detected in miloDE "showed signs" of DE (even though not detected while using corrected across neighborhoods $p$-values < 0.1 criteria), for each gene we calculated in how many neighborhoods we observe raw $p$-value < 0.05 (we denote those as "weakly DE"). We then split the 1400 genes (detected as DE in pseudo-bulk but no significant neighborhoods in miloDE) into several bins based on how many neighborhoods are weakly DE and in how many cells we detect the expression (Additional file 3: Fig. S12C). We observe two main classes of genes: genes with a very low number of cells with expression (< 300, 0.15%) and no weakly DE neighborhoods, and genes with a rather high number of expressed cells (> 1000) and more than 2% of weakly DE neighborhoods. We also observe an intermediate class of genes with a small number of weakly DE neighborhoods and a low number of expressed cells (200–1000). Interestingly, genes with a very low number of expressed cells show considerably higher effect size, thus identifying a burden of sporadic expression as an important limitation in the pseudo-bulk approach. Finally, it is also possible that genes marking differentially abundant cell states (e.g., Spp1-high cells against Spp1-low cells) may falsely be detected as DE while using a pseudo-bulk approach. To quantify the "DA marking potential" for each gene, we split cells into Spp1-high and Spp1-low populations, and performed pseudo-bulk DE between these groups, separately for healthy and disease cells (Additional file 3: Fig. S12D, Methods). Therefore, genes with higher absolute logFC in one or both tests are likely to be DE between DA sub-regions. As before, we split genes detected as DE in the pseudo-bulk analyses into two classes based on whether or not we detect any significantly DE neighborhoods in miloDE. However, we do not observe that genes that are not identified in miloDE show higher "DA/Spp1-marking potential" (Additional file 3: Fig. S12E), suggesting that in this particular dataset, this factor does not contribute to the discrepancy between pseudo-bulk DE and miloDE. Combined, we suggest that both sporadic expression and limitation of testing for a low number of cells contribute to differences in DE detection between miloDE and pseudo-bulk. We want to emphasize that miloDE is not intended as a substitute for the standard pseudo-bulk approach. Instead,

a joint analysis using both levels of granularity provides the most comprehensive and in-depth read-out of the alternative regulation of the disease.

Finally, to systematically assess the relevance of each gene set along the phenotypic progression of fibrotic macrophages, we used the logFC-DA to combine neighborhoods into 50 bins, and calculated the average fraction of significant genes and average logFC (Fig. 4F). The first 2 sets contain genes that when combined, are "uniformly" DE (both for the average logFC and a fraction of genes being DE) along DA axes (Fig. 4F; Additional file 3: Fig. S13). Next, the IPF-3 gene set also contains genes that are systematically and strongly (in terms of effect size) DE across all macrophages, with the effect size steadily increasing with the disease progression. This set contains a few of the known fibrotic macrophage markers such as *Fn1*, *Mmp7*, *Mmp10*, *Ppbp*, *Il1rn and Spp1* [8, 57]. Finally, IPF-4 and IPF-5 contain genes that are DE specifically in the later stages of the disease (particularly IPF-5), with IPF-4 containing some of the other known fibrotic markers such as *Csf1*, *Ctsk*, *Gpc4*, *Mmp9*, and *Sparc*.

When we perform functional annotation of the gene sets (Fig. 4G), we observe the enrichment of different biological processes in different gene sets, with the overall agreement with the previously suggested functional "timeline" of macrophage activation. IPF-2 set shows strong enrichment for genes associated with lipid biosynthetic process (*Cyp51a1*, *Msmo1*, *Dhcr24*) and "initial" antibacterial defense (*Defb1*, *Wfdc2*). Lipid metabolic reprogramming is suggested to play a role in the onset of IPF, and targeting lipid metabolism is potentially a relevant therapeutic strategy for IPF [60]. In addition, the response against pathogens is consistent with the observation that pathogenic bacteria are one of the drivers of IPF progression [61, 62]. Interestingly, although IPF-1 and IPF-3 exhibit distinct patterns along the DA axis, they "share" most of the enriched pathways, including some of the hallmarks of early and mid fibrosis. Thus, both IPF-1 and IPF-3 contain genes associated with inflammatory response and chemokine signaling (*Ccl23*, *Ccl4*, *Cybb* in IPF-1 and *Ccl24*, *Il1rn*, *Ccl7*, and *Ppbp* in IPF-3). We also detect enrichment for neutrophil degranulation (*Plac8*, *Itgam*, *Cybb*, *S100P*, and *Retn* for IPF-1 and *Chit1*, *Clec5a*, *Chi3l1* for IPF-3), consistent with the notion that neutrophils are among the first immune cells to be recruited for the site of inflammation [63]. Moreover, *Il1rn* and chemokines *Ccl23*, *Ccl4*, and *Ccl24* are associated with cellular response to interleukin-1, coherent with the observation that classically activated macrophages release IL-1beta chemokines [55, 64]. In contrast to the first 3 gene sets, IPF-4 and IPF-5 are associated with the more fibrotic phenotypes, specific to later stages. As mentioned previously, most of the previously suggested fibrotic markers belong to IPF-4, and consistent with this, IPF-4 is associated with the hallmark fibrotic processes such as platelet degranulation (*Sparc*, *Timp3*, *A2M*), extracellular matrix disassembly (*Mmp9*, *Ctsk*) and macrophage differentiation (*Csf1*, *Mmp9*). Finally, IPF-5 contains genes marking pathogenic, late-stage fibrosis such as wound healing (*Rhoc*, *Arhgap24*) and collagen fibril organization (*Col4a2*, *Itga6*, *Rhoc*). Combined, we illustrate that miloDE enables the discovery of subtle differences in gene regulation, with potentially important implications for how scRNA-seq data can be used to study the early stages of disease.

## Discussion and conclusions

The amount and diversity of scRNA-seq data is revealing ever more complex and subtle patterns that characterize processes ranging from normal development through to the onset and progression of disease. While broadly applied clustering strategies can provide high-level annotation of the heterogeneity within individual organs and tissues, their discrete nature is suboptimal for analyzing more granular changes in expression, including in the context of continuous trajectories. miloDE addresses this challenge in the context of testing for differential expression. Leveraging overlapping cell neighborhoods overcomes the tedious, frequently inaccurate, and time-consuming step of clustering and cell type annotation. Consequently, miloDE allows an in-depth characterization of various DE patterns, both across the whole dataset and within individual cell types. Given the rapid advances of extensive cohort atlas studies and large-scale CRISPR screens, miloDE is a timely computational tool, which successfully leverages a large number of replicates to enable sensitive and specific DE detection, while accounting for complex experimental designs.

While miloDE approximates gene regulation as a more continuous function of the manifold, it is important to acknowledge that it leverages the same principle of cell grouping as standard per cell type approaches. Therefore, in cases where a manifold is comprised of fairly distinctive, homogenous cell types, high-resolution clustering coupled with a per bulk DE approach will likely yield very similar results to miloDE. Similarly, in the case of continuous trajectories, a per bulk approach combined with the inclusion of a continuous covariate representing a pseudotime (or, perhaps, a more complex and bespoke approach such as tradeSeq[65]) might also yield similar results and resolution to miloDE. Moreover, differential pseudotime analysis such as Lamian and others can provide the insights for how certain lineages are affected, both in abundance and expression, during the disease or perturbation [66, 67]. While powerful in some cases, the trajectory-based branch of methods assumes that the manifold is continuous. In reality, a single-cell manifold is typically comprised of an intricate and complex combination of discrete and continuous cell types, and, moreover, it is often unclear whether the annotated cell types contain further heterogeneity. Combined, we suggest that miloDE allows the user to overcome laborious and bespoke analysis for each individual cell type using an elegant and straightforward approach.

Of note, we want to highlight two scenarios in which we believe miloDE, coupled together with Milo, is a powerful discovery tool. The first scenario is exemplified in our Tal1− analysis and it represents cases in which a disruption of the developmental process leads to a severe DA and depletion of cells of a certain cell type (which can be detected by Milo). We suggest that such depletion is often downstream of more subtle transcriptional changes that can be captured with miloDE. In addition, in clinical contexts, by coupling the analysis of DE and DA, it is possible to build a functional timeline for the disease progression and identify potential drug targets that are likely to be alternatively regulated early in the disease. Overall, we suggest that joint analysis by Milo and miloDE provides a powerful framework to characterize transcriptional changes and describe biological processes that are happening in the cells upon a perturbation.

Since miloDE leverages the principle of cell grouping, it is prone to similar pitfalls as standard per cell type approaches. To enable sensitive DE detection, we require

that neighborhoods exceed a certain size, which in turn can compromise the homogeneity of the cells in the neighborhoods. Within miloDE, we address this by introducing a 2nd-order kNN graph representation that provides an improved estimation for the local graph density. However, given the target number of cells per neighborhood, it is impossible to guarantee that transcriptionally distinct cells will not get assigned to the same neighborhood. This challenge can be partially mitigated by focusing on genes with strong effect sizes and by decreasing $k$.

Another important limitation that stems from using overlapping neighborhoods is the potential propagation of false positives. Specifically, while this approach overcomes limitations associated with discrete clustering, single cell(s) with an altered transcriptional makeup can dominate DE results in multiple tested groups. In other words, if even one cell has an extremely high (potentially sporadic) expression of a certain gene, some of the tested groups including this cell might be indicated as DE for this sporadic gene. Importantly, the issue scales with the increase in $k$ (which results in a higher degree of overlap between the neighborhoods), but it is also partially relieved by the neighborhood refinement step and can further be controlled by using a larger number of biological replicates. Another inevitable source of false positives is the extensive number of tests we perform—while the relative number of false positives (after multiple testing correction) is likely to be low (Fig. 2), the absolute number can still be substantial. Importantly, while individual gene-neighborhood combinations might result in false identification of DE and will require a manual examination and orthogonal verification, we suggest that the true exploratory power of miloDE lies in its ability to identify subtle and local patterns of co-regulated genes (thus minimizing the input from random FPs).

In this work, we adapted two readily available approaches for identifying DE-patterns—WGCNA and Louvain clustering. While they both have proved useful and have aided in biological discoveries, it is important to highlight their limitations. The WGCNA-based approach is an exclusive process meaning that only a fraction of genes will be assigned to any module, and therefore it is possible that interesting DE patterns will be excluded. In addition, the WGCNA-adapted approach to detect co-regulated gene modules is highly sensitive to input data, thus impeding the interpretation of the modules. On the other hand, the Louvain-adapted approach is inclusive of all input genes. However, clustering can be driven by a handful of neighborhoods, with the rest of the manifold being incoherently DE within individual gene clusters, thus rendering the characterization of the clusters challenging. Nonetheless, the combination of the suite of analysis methods allows for a flexible and tailored interpretation of the miloDE output.

Finally, moving forward, we suggest that in parallel with the emergence of novel cluster-free computational approaches for comparative analysis, it is crucial to contemplate how we will handle and interpret such complex outputs. One, nearly philosophical, conundrum is the decoupling between differential abundance and expression. While a mild change in expression for a certain gene in a certain transcriptional region can be detected as DE, it is possible that a stronger change in the expression will lead to the transition between transcriptional changes (i.e., DA). However, this depends on whether the gene in question contributed to the embedding in

the first place. miloDE is uniquely able to analyze DE and DA in parallel by combining the original Milo approach with miloDE. In sum, we believe that miloDE is a good stepping stone toward continuous comparative analysis, which is where the field of single-cell analysis is heading.

## Methods

### Code availability

Scripts to generate data and to perform the above analysis are available at https://github.com/MarioniLab/miloDE_analysis. It also contains the folder session_info listing all package versions. R Version is 4.2.3.

### miloDE pipeline

miloDE is a cluster-free Differential Expression (DE) framework that leverages a graph representation of single-cell data. In the first step of the pipeline, miloDE performs the assignment of cells to (overlapping) neighborhoods; each neighborhood contains neighboring cells, estimated from the graph representation. Once neighborhoods are assigned, DE testing is carried out for each neighborhood individually, and for each neighborhood and gene that was tested in this neighborhood, we return a logFC and *p*-value. Once the testing is concluded, we perform multiple testing correction in two directions: for each gene, across tested neighborhoods, and for each neighborhood, across tested genes. The final output consists of 4 [gene × neighborhood] matrices, containing logFC estimated, uncorrected *p*-value, corrected *p*-value across neighborhoods, and corrected *p*-value across genes.

Below we provide more details for each step:

1. *Neighborhood assignment*

    1. *Input*

        As input, the algorithm takes a SingleCellExperiment object [68] containing a count matrix for all the samples combined. We require that the colData slot contains entries that identify sample IDs (used as replicates in DE testing) and conditions to be tested. We also require that a pre-calculated joint latent embedding is provided in the reducedDim slot.

    2. *Graph construction and neighborhood assignment*
        We use 1st- or 2nd-order kNN graphs to represent transcriptional relationships between cells and assign neighborhoods (the order of the graph is specified by the user, default is 2). We define a 1st-order kNN graph as a standard kNN-graph, and the 2nd-order kNN graph as a 1st-order kNN graph augmented with edges between cells if they share at least one cell within their neighbors. Neighborhoods with the center cell *c* are defined as all cells (including *c*) that are connected with it by the edge. To define neighborhoods on a graph representation, we first select index cells (or cells that we will use as neighborhood centers in order to assign neighborhoods). Since the computational complexity of identifying index cells scales with the average density of the graph, and the 2nd-order

graphs on average are anticipated to be dense, we perform graph assignment and search for the index cells in parallel. Specifically, neighborhood assignment procedure goes as follows:

1.   We construct 1st-order (i.e., standard) kNN graph on all the cells, with $k = \min(50, k)$, where k is an input of the algorithm and has to be provided by the user.

2.   We then use this graph to select index cells using the waypoint sampling algorithm used in Milo2.0 [30, 69–71]. Specifically, we first select a random subset of cells, and the proportion of selected cells is defined by the parameter *prop* during the call of the corresponding function (default $= 0.2$). Then for each randomly selected cell ($c$), we extract the induced subgraph that consists of $c$ and all the neighbors of cell $c$(and all the edges between the vertices of the subgraph). We then calculate the number of 3-step cyclic random walks (triangles), and we select the vertex with the highest number of triangles as an index cell. Note that the number of selected index cells might be smaller than the number of initially randomly selected cells since following the described procedure, we can select the same index cell for several initially selected cells. This procedure allows to define a representative set of cells in the manifold rather than a simple random selection of the cells. Index cells are recorded, and we will use them later (step 5) as the neighborhood centers.

3.   Once index cells are selected, we recalculate the 1st-order kNN graph on all the cells if $k > 50$. If $k <= 50$, we use 1st-order kNN graph constructed in the 1st step. We note that implement step 1 first to create a "shallow" graph (i.e., with low $k$) on all of the cells that allows for fast implementation of step 2.

4.   If *order* $= 2$, we have an additional step 4 in which add edges between vertices of the graph from step 3 if they share at least one vertex as their neighbor.

5.   Once the graph is constructed (graph from step 3 for the 1st-order kNN or step 4 for the 2nd-order kNN), for each index cell selected in the 2nd step, we assign all its neighbors (and index cell itself) into a single neighborhood. In this way, each neighborhood is bijectively associated with its index cell.

6.   As an output of this step, we return the [cell × neighborhood] matrix with boolean identifiers of cell inclusion in the neighborhood. Note that cells, including index cells, might belong to several different neighborhoods. It is also possible that some cells are not "assigned" (i.e., none of the assigned neighborhoods contain the cell).

3.   *Neighborhood refinement*
     Graph refinement is optional, but a recommended step, during which we identify neighborhoods that can be discarded without any cells being unassigned to at least one remaining neighborhood. The graph refinement process represents a case of the NP-hard "set cover problem." Accordingly, we use a heuristic greedy implementation to solve the cover problem provided by the Rcpp::greedySetCover function [72, 73]. Within this implementation, we use neighborhoods as sets and cells as elements that can belong to the sets (i.e., a given cell is included in a given neighborhood). In this implementation of "set cover problem," sets are first sorted in decreasing order of their power (i.e., how many elements they contain), and then iteratively a set is included if it includes

at least one element (i.e., cell) that does not belong to any previously included sets. In the context of miloDE neighborhood refinement, we first assign an excessive number of neighborhoods to maximize the probability that all cells are assigned to at least one neighborhood; we use neighborhoods as sets and cells as elements.

2.  *DE testing*

1.  *Selection of neighborhoods to be tested*

As an optional step of the algorithm, we perform a selection of neighborhoods that show signs of "expression shifts" and therefore are recommended to be tested for DE. Discarding "unperturbed" (i.e., with no signs of "expression shifts") neighborhoods is computationally beneficial for downstream steps of the pipeline as well as facilitating the burden of multiple testing correction. To identify potentially "perturbed" neighborhoods, we adapt Augur, an approach that was originally designed to rank cell types by the degree of their perturbation in a specified condition [74].

Specifically, for each cell type, Augur builds a Random Forest classifier that predicts the condition to which a cell belongs, and for each cell type returns AUCs. To adapt Augur in the context of miloDE, for each neighborhood we return the AUC from the classifiers implemented by Augur. The user can then select their own AUC threshold to decide which neighborhoods should be supplied further for DE testing. We performed simulations and assessed how AUC distribution depends on the existence of DE between cell groups (Additional file 1: Supplementary Note 3). Note that our simulations show that in the presence of unbalanced batch effects nearly all neighborhoods, regardless of whether simulations contained DE genes or not, had AUC > 0.5 (which is a recommended default value for the AUC threshold). Since this step can be time-consuming, we recommend including it only in cases where the batch effect is anticipated to be minimal.

2.  *Selection of genes to be tested*
Following the same rationale of a priori discarding uninformative tests to reduce computing time and facilitate the burden of multiple testing correction, we provide an option to select genes to be tested. To do so, we employ edgeR::filterByExpr to determine which genes have sufficient counts to be considered for DE testing. The user can tune the minimum count required for at least some samples by changing the min_count parameter (default is 3). This procedure is performed for each neighborhood separately, which might result in different sets of genes being tested within each neighborhood. Note that min_count can be set to 0, and in this case, no gene selection will be performed.

3.  *DE testing within neighborhood*
To carry out DE testing, we use quasi-likelihood testing from edgeR, which was originally designed to perform DE testing on bulk RNA-seq. Specifically, to perform DE within each neighborhood, the algorithm proceeds as follows:

- •      We filter out all samples that contain less than min_n_cells_per_sample parameter, specified by the user (default is 3).
- • For each sample present in the neighborhood (i.e., biological replicate), we aggregate counts to create pseudo-bulks that mimic bulk RNA-seq data.
- • We select genes as described above, and using total sum counts across selected genes as a proxy for library size, calculate offset factors (edgeR::calcNormFactors) to correct for compositional biases.
- • We use the experimental design provided by the user. We allow the incorporation of covariates that are stored in the metadata of the count matrix.
- • We estimate the Negative Binomial dispersions as a function of gene abundance (edgeR::estimateDisp).
- • We estimate quasi-likelihood dispersions (edgeR::glmQLFit). The quasi-likelihood dispersion models variability of the per gene estimated variances.
- • We perform DE testing using a generalized linear model (edgeR::glmQLFTest). As an output, we get a table, where rows correspond to the tested genes, and columns contain estimated logFC, *p*-value, and FDR (i.e., *p*-values corrected across tested genes).

4.   *Multiple testing correction across neighborhoods*

Once tests for each neighborhood are performed, for each gene we correct calculated *p*-values across the tested neighborhoods. To do so, we adapt the spatial correction approach originally introduced in CYDAR [75] and further developed in Milo [30]. Specifically, we leverage the graph representation used to assign cells to neighborhoods, and apply the weighted Benjamini-Hochberg correction approach, where *p*-values for each neighborhood are weighted by the reciprocal of their local density. As a proxy for the local density for each neighborhood, we use a weighted sum across all cells in the neighborhood, where weights are calculated as the number of neighborhoods to which a cell belongs. Note that graph-based estimation for neighborhood density was developed after the original Milo publication and is available in the Bioconductor version 3.16 (BiocManager::install(version = "3.16")). Finally, we only perform the correction across the neighborhoods for which testing for the gene was carried out. For neighborhoods for which testing was not performed, we return NaN.

Above we described each step of the pipeline. To facilitate the appropriate use of the algorithm, we combined the recommendations for the optimal selection of the hyperparameters of the algorithm (Additional file 3: Fig. S14 and Additional file 4: Table S2). We suggest that the most appropriate parameters to consider are the different order-k combination which define the distribution of neighborhood sizes and thus impact the sensitivity of DE detection. Additional file 4: Table S2 provides a reference table that allows DE to be estimated as a function of effect size, number of tested cells and replicates. Additionally, to estimate how neighborhood size distribution depends on order-k for any specific dataset, a corresponding function miloDE::estimate_neighbourhood_sizes is available within the R package.

**Analysis of interplay between differential abundance (DA) and differential expression (DE) in "unsupervised" and "supervised" embedding schemes using simulations**

1. *Base simulation*

   To generate base simulation (i.e., no DE and no DA between the conditions), we used the R package splatter that simulates scRNA-seq counts with the desired properties [76]. To estimate parameters for the simulations, we used WT mouse embryo data from a single sample of developmental stage E8.5. We restricted the analysis to 1000 highly variable genes (parameters are listed in Additional file 6: Table S4). To simulate two cell types, we simulated two cell groups with 10% of genes being DE. Henceforth we refer to these genes as genes contributing to shared variance or "shared" genes. In total, we simulated 4 batches (i.e., replicas), and split batches in half, with one half being assigned as condition A and another half assigned as condition B. Finally, for genes that are originally DE in the base simulation (i.e., genes that contribute to the difference between the cell types), we calculated the average expression level in cell type 1 and cell type 2, and we discarded genes with absolute difference between average levels of expression lower than 0.5. To calculate log-normalized counts, we used scuttle::logNormCounts [77].

2. *"Perturbed" simulations*

   To generate perturbed simulations, in which we introduce DE and/or DA between two conditions, we altered counts for cells from cell type 1 and condition B. We controlled this synthetic perturbation such that altered cells from cell type 1 will acquire a transcription profile more similar to those from cell type 2. Specifically, we varied the number of altered "not shared" genes as well as the magnitude of the expression shift for all "shared" genes. To select an appropriate shift for "shared" genes (and thus enable cells from cell type 1 to "move" toward cells from cell type 2), for each "shared" gene we calculated the median of expression in cell type 1 and multiplied it by the effect size of their initial DE (i.e., the difference between cell type 2 and cell type 1) multiplied by the magnitude of the shift (a variable that we varied between 0 and 1). This calculated value is referred to as range, and for each "perturbed" cell we manually added random counts, sampled from the vector c(0, range). Intuitively, if the magnitude of the shift is 0, then no perturbation is applied. On the other hand, if the magnitude of the shift is 1, most cells from cell type 1 acquire an average phenotype of cells from cell type 2. For a magnitude of the shift between 0 and 1, perturbed cells will acquire an intermediate phenotype between cell type 1 and cell type 2. To alter "not shared" genes, for each "not shared" gene, we calculated its range as its base expression in cell type 1 multiplied by 0.5. In total we used a 2-dimensional grid: for the magnitude of shift for all "shared" genes, we used a grid c(0, 0.02, 0.05, 0.1, 0.15, 0.2, 0.4, 0.5, 0.75, 1), and for the number of altered "not shared" genes we used a grid c(0, 2, 5, 10, 15, 20, 30, 40, 50, 100). To calculate log-normalized counts, we used scuttle::logNormCounts.

3. *"Supervised" and "unsupervised" embeddings*

To perform "supervised" and "unsupervised" embedding in this controlled setting, we calculated principal component analysis (henceforth PCA) either only on "shared" genes (for the "supervised" embedding) or on all genes ("unsupervised" embedding).

4. *Neighborhood assignment*

   For each dataset and embedding, we generated a kNN graph ($k = 100$, $order = 1$) and assigned cells to neighborhoods. We then discarded redundant neighborhoods as described above. Additionally, to assess whether the difference between embedding approaches and perturbations is robust to neighborhood assignment, we performed 3 independent neighborhood assignments.

5. *DA analysis*

   We used Milo [30] to assess the degree of DA in each dataset, neighborhood assignment, and embedding type. We then calculated the fraction of neighborhoods with SpatialFDR lower than 0.1.

6. *DE analysis*

   We used miloDE to assess the degree of DE in each dataset, neighborhood assignment, and embedding type. We then calculated the average fraction (across all perturbed "not shared" genes) of neighborhoods with a corrected across neighborhoods $p$-value $< 0.1$.

### Analysis of the effect of embedding approaches on miloDE performance

1. *Dataset*

   We use the mouse gastrulation scRNA-seq data from [41]; data downloaded using the MouseGastrulationData() package (Griffiths and Lun, 2022). As WT cells, we use all samples from the E8.5 developmental stage. As ChimeraWT cells, we use all cells from the WT chimera (i.e., no knock out is introduced) mouse embryos (which are also from developmental stage E8.5). Next, we concatenated counts from both experiments, and embeddings were calculated on log-normalized counts (in a batch-aware manner, using batchelor::multiBatchNorm [34]).

2. *Embeddings*

   We analyze several embedding approaches, covering several widely used methods, both "unsupervised" and "supervised" settings, and different selections of highly variable genes (HVGs). Specifically, we used MNN [34], Azimuth [4], scVI [35] and scANVI [78]. We also implemented a "supervised" version of MNN, and for the "supervised" versions of scVI and scANVI, we used scArches [5]. For the "unsupervised" version of Azimuth we used only its first step of anchor-based integration [79] while using all the samples regardless of whether they are coming from reference or query condition. To select HVGs, we used either reference data or the union of reference and query data. All approaches perform batch-aware embedding, and sample ID was used as the batch. For all embeddings, except scVI and scANVI, we calculate a 30-dimensional latent space. For scVI and scANVI, we use the provided default (10). Below we introduce more detailed descriptions for each approach.

   - MNN is an "unsupervised" approach that performs batch-aware PCA correction followed by MNN correction. The MNN-based latent space is sensitive to the

selected genes for which we perform PCA. Accordingly, we select 3000 highly variable genes using scran::getTopHVGs [80], and the selection is performed using either only WT data; only ChimeraWT data; WT + ChimeraWT data.

- Reference-projected MNN is a "supervised" variation of the original MNN. Specifically, we first perform batch-aware PCA on control data using batchelor::multiBatchPCA. As an output, we get batch-corrected PCs for the control data as well as the matrix of rotation vectors that we can apply to calculate PCA on scaled case data. Finally, we concatenate the calculated PCs and perform MNN correction using batchelor::reducedMNN.
- Azimuth integration follows the pipeline for the reference-based mapping introduced by [4] and is employed by the Azimuth web application for reference-based single-cell analysis (https://azimuth.hubmapconsortium.org/). To perform integration, we follow the provided by the Azimuth mapping pipeline.
- scArches integration follows the pipeline from [5].

3. *Chimera-specific DE genes*
   To identify chimera-specific genes that are systematically upregulated across all cell types, we performed DE testing for each cell type and selected genes that have negative logFC (i.e., upregulated in ChimeraWT) and are significantly DE (FDR < 0.1) in at least 75% of all cell types. We then selected 3 genes with a variable base expression (i.e., expression in reference cells).

4. *Neighborhood assignment and estimation of neighborhood homogeneity.* For each embedding, we generated a kNN graph ($k = 100$, *order* = 1) and assigned cells to neighborhoods. We then discarded redundant neighborhoods as described above. Additionally, to assess whether the difference between embedding approaches is robust to stochasticity in neighborhood assignments, we performed 5 independent neighborhood assignments. To assess the homogeneity of the neighborhoods, for each embedding and each chimera-specific DE gene, we calculated the standard deviation of log-normalized counts within each neighborhood across ChimeraWT cells.

5. *Estimation of logFC for chimera-specific DE genes.*
   To estimate logFC distribution for each embedding and each chimera-specific DE gene, we performed miloDE for each embedding (neighborhood assignment based on kNN graph, $k = 100$) and aggregated logFC estimates across all neighborhoods.

6. *Estimation of the fraction of significantly DE neighborhoods for chimera-specific DE genes*
   To estimate the fraction of significantly DE neighborhoods, for each embedding and each chimera-specific DE gene, we calculated the fraction of neighborhoods with corrected (across neighborhoods) *p*-value lower than 0.1.

## Simulations to estimate the relationship between the number of tested cells and DE detection

To assess how DE detection using quasi-likelihood testing from edgeR depends on the number of cells, we used the R package splatter that simulates scRNA-seq counts with the desired properties [76]. To estimate parameters for the simulations, we used wild type (WT) mouse embryo data from a single sample of developmental stage E8.5.

Additionally, we restricted the analysis to 4000 highly variable genes (parameters are listed in Additional file 6: Table S4). We then simulated the two main datasets (one without batch effect between replicates and one with), which contained two cell groups (imitating control and case samples, the probability to be assigned in each group is 50%) in which we assigned 25% of genes to be DE (mean logarithm of the effect size was assigned to 1). Additionally, we required both cell groups to have 10 batches (i.e., replicates). To estimate how the number of replicates affects the detection, we subsampled 5 datasets from each of the two main datasets (with and without batch effect), by randomly selecting either 2, 4, 6, 8 or 10 replicates per each condition. In addition, to estimate how the imbalance of the number of replicates between control and case groups affects the detection, we subsampled 7 datasets from the main dataset without batch effect, by keeping the total number of replicates across both conditions as 12 and randomly selecting either 2, 3, 4, 6, 8, 9 or 10 replicates from the control condition (and, respectively, selecting 10, 9, 8, 6, 4, 3, 2 random replicates from the case condition). For each subsampled dataset, we then subsampled a random number of cells (higher than 50 and lower than min(3000, number of cells in the dataset)) and estimated DE detection for the selected cells (all genes that had $p$-value < 0.05 were assigned as detected). For each subsampled dataset, we downsampled cells 2000 times, and to estimate the overall trend between the number of cells and DE detection parameters (sensitivity, specificity, FDR), we calculated the running median across 2000 downsamplings. Finally, to assess how DE detection scales with the effect size, we performed a similar procedure, but when varying the meanlog parameter in range c(1, 2, 3).

### Estimation of the relationship between the assignment (i.e., order-k) and neighborhood size distribution

To estimate how neighborhood size distribution depends on the neighborhood assignment, we used WT mouse embryo data (developmental stage E8.5; using the provided MNN-corrected PCs as latent space) and for a wide range of [*order-k*] combinations performed neighborhood assignment. Specifically, for $order = 1$, we used $k = \mathrm{seq}(50, 500, 50)$, and for $order = 2$, we used $k = \mathrm{seq}(5, 50, 5)$.

### Analysis of how different orders affect the sensitivity of DE detection

1. *Dataset*

   We used mouse gastrulation scRNA-seq data from [41]. As WT cells (control), we use all samples from the development stage E8.5. As ChimeraWT cells (case), we use tD-tomato negative cells from Tal1− chimera mouse embryos (which are also assigned with developmental stage E8.5). Accordingly, we concatenated counts from both experiments, and embeddings were calculated on log-normalized counts (in a batch-aware manner, using batchelor::multiBatchNorm). For the latent embedding, we used scArches integration.

2. *Selection of cell types*
   We selected all cell types with at least 20 cells in each condition.

3. *Neighborhood assignments*

We used the following range of [*order-k*] combinations to assign cells to neighborhoods: for $order = 1$, $k = $ c(75, 150, 200, 250, 300, 350, 400, 450); for $order = 2$, $k = $ c(10, 15, 20, 25, 30, 35, 40, 45). For each [*order-k*] combination, we performed 10 separate neighborhood assignments.

4. *Per neighborhood estimation of cell type purity*

   For any given cell type, we calculated a per neighborhood cell type purity score as the fraction of cells from the neighborhood annotated with the cell type (relative to the total number of cells in the neighborhood). For any given neighborhood assignment, we then calculated the maximum cell type purity score (across all neighborhoods from the assignment) as a proxy of how specifically the assignment results in grouping the cells from the cell type in question.

5. *Per neighborhood estimation of relative cell type enrichment*

   To calculate the relative cell type enrichment score, we first annotated each neighborhood with its most abundant cell type, and then for each neighborhood we calculated the fraction of cells that are annotated as being from that cell type.

6. *Simulation of counts to supply ground truth DE*

   To simulate differences in counts in a targeted manner, for each of the selected cell types, we first identified candidate genes that are not DE between the conditions across all tested cell types. For each cell type, we then selected 60 "candidate" genes and added a random integer number of counts (from 0 to 2) to cells from the control condition in the cell type. We note that for some pairs of cell types we selected the same gene(s), and in these cases, we augmented the count matrix with copies of the gene(s) and treated each gene copy as a separate copy per selected cell type. In addition, to ensure a wide range for logFC, we ensured a varying base expression for the selected genes. Finally, we estimated logFC for each gene/cell type and focused the analysis on genes for which the absolute estimated logFC varies between 1 and 6 (45 genes per cell type on average).

7. *Analysis of how sensitivity in DE detection depends on cell type purity and the absolute number of cells from the cell type*

   We applied miloDE for each neighborhood assignment and, for each neighborhood and "perturbed" cell type, we estimated its cell type purity, the absolute number of cells from the cell type, and DE detection power (i.e., fraction of associated "perturbed" genes that show *p*-value < 0.05).

**Analysis of the comparison between standard and refined neighborhood assignments**

To assess whether refined neighborhood assignment minimizes the number of neighborhoods while ensuring complete coverage of cells with at least one neighborhood, we used WT mouse embryo data (developmental stage E8.5; using the provided MNN-corrected PCs as latent space). We fixed $order = 2$ and for $k = $ seq(10, 50, 10) performed neighborhood assignments, followed by the refinement step (for each $k$, we performed the procedure 5 times). Then, for each neighborhood assignment, we calculated the total number of neighborhoods and performed matched (for the number of neighborhoods) assignments without the refinement step. For each comparison (i.e., each $k$), we calculated the fraction of cells that did not get assigned to any neighborhoods.

**Analysis of how AUC distribution from Augur-based classifiers depends on DE**

To assess how the AUC distribution depends on whether DE is present between two tested groups, we used the R package splatter to simulate scRNA-seq counts [76]. To estimate parameters for the simulations, we used WT mouse embryo data from a single sample of developmental stage E8.5. Additionally, we restricted the analysis to 4000 highly variable genes (parameters are listed in Additional file 6: Table S4). We then simulated several datasets (54 in total), containing two tested conditions; we used 5 replicates for each condition:

- Fraction of DE genes $= 0$, 5 or 25%.
- Fraction of cells in control condition $= 25$, 50 or 75%.
- Effect size (de_facLoc) $= 1,2$.
- Without batch effect, with balanced batch effect (identical for one case–control replicate pair), and with unbalanced batch effect. For the unbalanced batch effect, we subsetted datasets with a balanced batch effect by randomly discarding 2 case and 2 control replicates.

For each dataset, we then randomly sampled cells 100 times (ensuring that the sample size lies between 75 and 500), and for each subsampling, we calculated the AUC for the classifiers that separate two conditions.

**Analysis of the performance of miloDE on simulated data**

1. *Dataset*

   We utilized the mouse gastrulation scRNA-seq data from [41]. As WT cells, we considered all samples from the E8.5 development stage. As ChimeraWT cells, we use tD-tomato negative cells from Tal1− chimera mouse embryos (which are also sampled from E8.5). We concatenated counts from both experiments, and embeddings were calculated on log-normalized counts (in a batch-aware manner, using batchelor::multiBatchNorm). For the latent embedding, we used scArches.

2. *Simulation of counts to supply ground truth DE*
   We first sub-clustered cells annotated as Forebrain/Midbrain/Hindbrain using Louvain clustering, and based on the expression of known marker genes *Shh, Rax,Six3, Otx2, En1, Hoxb2, Hoxa2, Gbx2* [81–86], we annotated brain sub-clusters with one of the following (sub)cell types: Forebrain, Midbrain, Hindbrain and Floor plate. To simulate differences in counts in a targeted manner, we first selected a sub-cell type (Floor plate) in which to perturb the counts for several genes. We then identified candidate genes as genes that are not DE between the conditions (i.e., not DE within all abundant cell types (number of cells > 50)), and then selected 50 genes and added a random integer number of counts (from 0 to 2) to cells from the control condition. In addition, to ensure a wide range of logFCs was tested, we considered genes with a wide range of base expression levels. Finally, we randomly selected 5 genes with varying effect sizes, ranging from 1 to 4.5.

3. *Assessment of DE detection per neighborhood assignment, cell type purity threshold, and "perturbed" gene*

Within each neighborhood assignment, we calculated the cell type purity score per neighborhood as the fraction of cells in each neighborhood annotated as being from the cell type of interest. Accordingly, for each cell type purity threshold, all neighborhoods with cell type purity exceeding the designated threshold are annotated as "ground truth" DE. Next, for each "perturbed" gene we identified neighborhoods as DE if corrected across neighborhoods *p*-value was less than 0.1. We then assessed DE detection power by estimating sensitivity and FDR.

### Comparison of the performance between miloDE and Cacoa

1. *Simulation of the base datasets*

We used the R package splatter to simulate scRNA-seq counts [76]. To estimate parameters for the simulations, we used wild type (WT) mouse embryo data from a single sample of developmental stage E8.5. Additionally, we restricted the analysis to 3000 highly variable genes (parameters are listed in Additional file 6: Table S4). Since the goal of the benchmark is to compare the ability of both methods to sensitively and accurately detect DE in a subset of cells (imitating DE restricted to a sub-cell type), we simulated several datasets, where each contained varying fractions of two groups (i.e., two sub-cell types). Specifically, we simulated seven datasets, where the fraction of the group to which we later will "apply the perturbation" (we denote it as perturbed group), varied in the range c(0.005, 0.01, 0.025, 0.05, 0.1, 0.25, 0.5). For each value from this range, we manually adjusted the number of DE (between groups) genes and the effect size in such a way, that the simulated groups are not entirely separated in the latent space while using graph representation and neighborhood assignment from miloDE (i.e., there are neighborhoods that contain cells from both groups). For each dataset, we generated 5 batches (i.e., replicates) per condition. To assess how the performance scales with the number of replicates per condition, for each dataset, in addition to the dataset itself, we generated 3 downsamplings, by restricting the number of batches per case or condition or both to 2. In total, we generated 28 datasets, with varying fractions of perturbed group and numbers of replicates. To construct a latent space, we used MNN-corrected PCs (10 PCs).

2. *Simulation of counts to supply ground truth DE*

For each dataset, we first identified candidate genes as genes that are not DE between two simulated groups. For each dataset, we then selected 200 "candidate" genes and added a random integer number of counts (from 0 to 3) to cells from the case condition in the perturbed group. We then selected 9 genes for which we estimated the expected effect size (by pseudo-bulk comparison between the conditions for the perturbed group) to lie in range c(1, 1.5, 2, 2.5, 3, 3.5, 4, 4.5, 5).

3. *miloDE*

We applied miloDE in 15 neighborhood assignments (*k* varying in range c(20, 25, 30), and to assess robustness across different assignments, for each *k* we performed 5 independent neighborhood assignments). Additionally, since we preselected 3000

variable genes for the simulation, we removed the requirement of having a minimum expression level per neighborhood.

4. *Cacoa*

   We implemented Cacoa using the tutorial from the Github page. Similar to miloDE, Cacoa also requires a graph representation to aggregate the information across neighbors. We generated such a graph representation using Seurat::FindNeighbors. For each dataset, we generated 4 graph representations, by varying $k$ for Seurat::FindNeighbors in the range c(25, 50, 100, 200).

5. *Comparison of the performance between miloDE and Cacoa*

   Since the statistical outputs of miloDE and Cacoa are not directly comparable ($p$-values for miloDE and $z$-score for Cacoa), and it is not entirely clear which threshold for $z$-score in Cacoa to use as a cut-off of significance, we used the AUC to compare performance. Moreover, since Cacoa estimates DE for each cell (in which ground truth is provided in the simulations), and miloDE estimates DE per neighborhood (for which ground truth is unknown and instead could be estimated with the threshold for the Group purity), we approached the estimation of AUC on the neighborhood and single-cell level. For the neighborhood-level estimation, we aggregated the $z$-scores across the neighborhoods assigned with miloDE (using the neighborhood assignments with $k = 25$, for each neighborhood we aggregated the average $z$-score (raw or adjusted) across all the cells from the neighborhood). We used two thresholds (0.1 and 0.25) for the perturbed group fraction to decide which neighborhoods contain enough cells to be considered positive. This allowed us to estimate the AUC for each approach, perturbed gene, and the designated "perturbed group fraction" threshold. For the single-cell-level comparison, we "deconvolved" the output of miloDE to return the statistic for each cell. To do so, we estimated a $z$-score for each cell from miloDE output, by first transforming $p$-values into $z$-scores for each neighborhood (using stats::qnorm, lower.tail = FALSE), and then, for each cell, taking the average $z$-score across all neighborhoods to which it was assigned. In this comparison, cells with ground truth DE are directly known from the simulations. By design, Cacoa returns a $z$-score equal to 0 for cell/gene combinations with no expression, and we further estimated Cacoa's performance using only case perturbed cells as ground truth DE or using all cells from the perturbed group as ground truth DE.

**Comparison of the performance between miloDE and *pseudo*-bulk approach**

We used the same datasets described earlier to benchmark miloDE against Cacoa. We evaluated the performance of miloDE and pseudo-bulk approach (edgeR) on a bigger set of genes, with the estimated effect size for the perturbed group from the range seq(1, 4, 0.25). Since the outputs between miloDE and the pseudo-bulk approach are not directly comparable, we estimated the sensitivity of the pseudo-bulk approach for the selected genes as a boolean (FDR < 0.1) whereas the sensitivity of miloDE for each gene was estimated across the neighborhoods. To do so, for each assignment and each group purity threshold, we estimated sensitivity as the number of true positive neighborhoods (i.e., neighborhoods purity is higher than a designated threshold and corrected $p$-value across neighborhoods < 0.1) over the number of all positive neighborhoods (neighborhoods purity is higher than a designated threshold).

**Comparing Tal1 + and Tal1 − embryonic cells**

1. *Dataset*

   We used mouse gastrulation scRNA-seq data from [41], specifically Tal1 chimera data, that can be loaded with Tal1ChimeraData(). We used tomato-td row to identify Tal1+ and Tal1− cells. We calculated log-normalized counts using scuttle::logNormCounts [77]. For the latent embedding, we first selected 3000 highly variable genes for Tal1+ cells, and then we used MNN on batch-corrected PCs on the selected genes.

2. *Cell type ranking by the extent of transcriptional shifts*
   To assess how different cell types are affected by the lack of Tal1, we first assigned neighborhoods across the whole dataset (*order = 2, k = 25*). We then retained only variable genes (by selecting genes with positive variance based on scran::modelGeneVar estimates), and applied miloDE testing; within each neighborhood, we tested only for genes that were expressed in at least some cells (using min_count = 3). To estimate the extent of the transcriptional shift for each neighborhood, we calculated two metrics: the number of DE genes (corrected across genes $p$-value < 0.1) and the number of "specifically" DE genes. To calculate the number of specifically DE genes, for each gene we first $z$-normalized corrected across the neighborhoods $p$-values. Accordingly, a gene-neighborhood combination is denoted as specifically DE, if its $z$-normalized $p$-value is below −3, and for each neighborhood we calculated the total number of genes with a $z$-normalized$p$-value < −3. Note, that for all gene-neighborhoods combinations that were not tested, we assigned $p$-value to 1. Finally, we assigned each neighborhood the most enriched cell type label across the neighborhood's cells, and for each cell type, we calculated the distribution (across corresponding neighborhoods) of the number of DE genes (total and neighborhood-specific).

3. *miloDE analysis of cells contributing to blood lineage*
   To systematically assess how the absence of Tal1 transcriptionally manifests in cells contributing to a blood lineage, we selected cells annotated as hematoendothelial progenitors or endothelium and assigned neighborhoods (*order = 2, k = 20*). We then retained only variable genes (by selecting genes with positive variance based on scran::modelGeneVar estimates), and applied miloDE testing; within each neighborhood we tested only for genes that were expressed in at least some cells (using min_count = 3).

4. *Per neighborhood estimation of the proximity to blood progenitors*
   For each cell annotated as hematoendothelial progenitors or endothelium, we calculated the minimum distance (in PC space) to cells that were annotated as blood progenitors. Accordingly, for each neighborhood, we calculated the average (across its cells) of these minimum distances.

5. *Identification of co-regulated gene modules*
   We adapted the WGCNA framework to identify gene modules consisting of co-regulated genes. Instead of expression vectors, we used logFC values (across neighborhoods). Additionally, to minimize the input from the neighborhoods that are not DE,

we assigned all logFC values to 0 if the corrected across neighborhoods *p*-value > 0.1. Additionally, for each gene-neighborhood combination that was not tested, we assigned logFC values to 0 and *p*-values to 1. We restricted our analysis to genes that are DE in at least 2 neighborhoods. Finally, to identify gene modules, we used the R package scWGCNA which is specifically tailored to handle scRNA-seq data [46]. Specifically, we employed scWGCNA::run.scWGCNA using neighborhoods instead of single cells (and therefore skipping the calculation of pseudocells), otherwise with the default settings.

6. *Gene ontology enrichment analysis*
   To assess which biological processes are enriched across different gene modules, we used enrichR::enrichr (within GO_Biological_Process_2021 database) [87, 88]. For each gene module, all gene ontologies with adjusted *p*-value < 0.1 were assigned as significantly enriched.

**Analysis of macrophage-specific transcriptional shifts upon idiopathic pulmonary fibrosis**

1. *Dataset*

   Four datasets, containing cells from both healthy and IPF donors [8, 57, 58, 89], were downloaded from the original source and mapped onto the Azimuth lung reference using scripts available at https://github.com/satijalab/azimuth-references/tree/master/human_lung. Azimuth lung reference consists of 65,662 human lung cells from (https://app.azimuth.hubmapconsortium.org/app/human-lung, [90]) (a processed dataset that is immediately suitable for the Azimuth mapping is available at https://zenodo.org/records/4895404).

2. *Neighborhood assignment and DE testing*
   To characterize macrophage-specific transcriptional shifts, we selected all cells annotated as macrophage, and assigned neighborhoods (*order*= 2, $k$ = 30). We then retained only variable genes (by selecting genes with positive variance based on scran::modelGeneVar estimates), and applied miloDE testing, using the dataset ID as a covariate. Within each neighborhood, we tested genes that are expressed in macrophage annotated cells (gene selection was performed based on the output of edgeR::filterByExpr, min_count = 3). Additionally, within each neighborhood we discarded donors, for whom we had less than 3 cells in the neighborhood.

3. *Identification of gene sets*
   To identify macrophage-specific gene sets, we clustered genes using Louvain clustering. We restricted the analysis to genes that were strongly upregulated in IPF donors. Specifically, we selected genes that satisfied the following criteria:

   • Corrected across neighborhoods *p*-value < 0.1 in at least 25% of the neighborhoods.
   • Absolute average logFC across significant (i.e., corrected across neighborhoods *p*-value < 0.1) neighborhoods > 1.
   • Gene is upregulated in IPF donors (i.e., has negative logFC) in at least 75% of neighborhoods.

As a vector for each gene, we use logFC (across neighborhoods), with logFC being set to 0 if the corrected across the neighborhoods *p*-value > 0.1 (henceforth referred to as corrected logFC). We then computed the shared nearest neighbors graph (on the first 5 PCs, using scran::buildSNNGraph) and Louvain clustering was calculated using igraph::cluster_louvain [59, 69] (resolution = 1).

4. *Per gene set discovery of "marker" neighborhoods*
To identify the neighborhoods in which genes between different gene sets are DE, we used scran::findMarkers on logFC vectors (across neighborhoods; using corrected logFC). We then selected the "top" 3 neighborhoods per gene set, where top neighborhoods were defined by the "Top" column in scran::findMarkers output and represent a minimal number of neighborhoods required to separate any cluster from any other cluster (specified by pval.type = "any").

5. *Characterization of DE patterns based on the prevalence of DE neighborhoods in different stages of fibrotic progression*
To characterize whether a gene is DE preferably in the neighborhoods that are significantly enriched for IPF cells or in the neighborhoods that contain a comparable number of cells from healthy and IPF donors, we restricted the analysis to neighborhoods with negative logFC according to the DA test and split them into two groups: neighborhoods that are significantly enriched for IPF cells (SpatialFDR for DA test < 0.1) and neighborhoods that contain a comparable number of cells from healthy and IPF donors (SpatialFDR for DA test ≥ 0.1). Then, for each gene, we extracted the number of DE neighborhoods (*p*-value corrected across neighborhoods < 0.1) in each of the groups and performed a Fisher test to assess whether any of the two neighborhood groups contain significantly more DE neighborhoods. We performed a Fisher test for all genes that we used for the Louvain clustering and used a corrected *p*-value (cutoff of 0.1) to decide whether the difference is significant or not. Accordingly, we split all genes into 3 classes: genes that are DE significantly more frequently in either of the two groups and genes that are DE relatively equally in both groups.

6. *Comparison with per cell type DE estimation*
To assess whether genes that we identify with miloDE as upregulated in IPF are also DE on a whole cell type level, we implemented the same edgeR framework but across all macrophages (we performed testing only for genes that we used for the clustering, i.e., genes that are strongly upregulated in IPF donors). We then assigned genes with FDR < 0.1 as significantly DE on the cell type level. We identify detection in miloDE as the existence of neighborhoods for which corrected across neighborhoods *p*-value < 0.1 and being "weakly DE" in miloDE is defined by the existence of neighborhoods with raw *p*-value < 0.05. To retrieve per-gene estimates of "DA-marking capacity," we split all cells into Spp1-low and Spp1-high cells (Spp1-low if normalized log-counts < 0.25), and we performed DE between two groups using pseudo-bulk edgeR. This procedure was done separately for healthy and disease cells, and donor identity was used as a covariate.

7. *Per gene set characterization of average DE pattern along the fibrosis progression*
To characterize per gene set DE patterns along the fibrosis progression, we grouped neighborhoods based on their logFC for the DA test into 50 equal-sized bins. For

each set and logFC-DA bin, we then calculated the average (across all aggregated neighborhoods and genes from the gene set) corrected logFC across all genes from the set and the average (across all aggregated neighborhoods) fraction of genes (from the gene set) that are significantly DE in the corresponding neighborhood.

8. *Gene ontology enrichment analysis*

    To assess which biological processes are enriched across different gene modules, we used enrichR::enrichr (within GO_Biological_Process_2021 database). For each gene set, we then selected the top 5 gene ontologies (based on adjusted *p*-value) and for the union of all top gene ontologies, we estimated how significantly each of them is enriched in each gene set.

## Supplementary Information

Additional file 1: Supplementary Notes.

Additional file 2: Table S1. Table with estimated statistics of DE detection as a function of the number of replicates, number of cells and effect size.

Additional file 3: Supplementary Figures.

Additional file 4: Table S2. List of enriched gene ontologies for each module detected in Tal1- analysis.

Additional file 5: Table S3. List of genes that are strongly DE in IPF (> 25% neighborhoods, average significant logFC > 1), with assigned gene set ID.

Additional file 6: Table S4. Specific parameters of splatter simulations.

Additional file 7: Peer review history

## Acknowledgements

We thank Mike Morgan and Alice Kluzer for optimizing and scaling the original Milo approach which enabled a scalable implementation of miloDE. We also thank Jean Fan, Brendan Miller, Austin Hartman, and Gesmira Molla for fruitful discussions that helped to shape the method and interpretation of the results. We thank Berthold Göttgens, Luke Harland, and Bart Theeuwes for the discussions and valuable feedback on the implementation of miloDE in the context of chimeric mouse embryos lacking Tal1. We thank Ana-Maria Cujba and Amanda Oliver for the discussions and valuable feedback on the implementation of miloDE in the context of macrophage activation in IPF. We thank Hector Corrada Bravo and Oleg Mayba for the discussion of the benchmark and the statistical aspects of edgeR testing.

## Review history

The review history is available as Additional file 7.

## Peer review information

## Authors' contributions

A.M., R.S., and J.C.M. conceived the method. A.M. developed the method, wrote the code, and performed the analysis, with input from E.D. and L.R.. All authors interpreted the results. A.M. and J.C.M. wrote the manuscript, with input from all the authors. J.C.M. and R.S. oversaw the project.

## Funding

 This work is supported by the National Institutes of Health: RS and JM acknowledge OT2OD026673 and OT2OD033760 which supports AM; RS acknowledges support from NIH (RM1HG011014-02) and Chan Zuckenberg Initiative (EOSS5-0000000381 and HCA-A-1704–01895); ED acknowledges Wellcome Sanger core funding (WT206194); LR is funded by the EMBL International PhD Programme and is a member of Darwin College of the University of Cambridge. JM acknowledges core funding from EMBL and core support from Cancer Research UK (C9545/A29580). In the past 3 years, RS has worked as a consultant for Bristol-Myers Squibb, Regeneron, and Kallyope, and served as an SAB member for ImmunAI, Resolve Biosciences, Nanostring, and the NYC Pandemic Response Lab. JM has been an employee of Genentech since September 2022.

## Availability of data and materials

miloDE's open-source code is maintained and documented on GitHub [91] and is publicly available under the MIT license from the following Zenodo repository (DOI: https://zenodo.org/records/12686748) [92].
The datasets that we used for the analysis are available:
Mouse embryo scRNA seq:
Berthold Gottgens, Blanca Pijuan-Sala. [Blanca Pijuan-Sala, Jonathan A Griffiths, Carolina Guibentif, Tom W Hiscock, Wajid Jawaid, Fernando J Calero-Nieto, Carla Mulas, Ximena Ibarra-Soria, Richard C V Tyser, Debbie Lee Lian Ho, Wolf Reik,

Shankar Srinivas, Benjamin D Simons, Jennifer Nichols, John C Marioni, Berthold Gottgens] A single-cell molecular map of mouse gastrulation and early organogenesis. Processed single cell expression data for WT mice and Tal1 − chimeras can be downloaded from here: https://git.bioconductor.org/packages/MouseGastrulationData. DOI: https://doi.org/10.1038/s41586-019-0933-9 (2019).

Four datasets, containing cells from both healthy and IPF donors [8, 57, 58, 89] were downloaded from the original source and mapped onto the Azimuth lung reference using scripts available at https://github.com/satijalab/azimuth-references/tree/master/human_lung. Azimuth lung reference consists of 65,662 human lung cells from (https://app.azimuth.hubmapconsortium.org/app/human-lung, [90]) (a processed dataset that is immediately suitable for the Azimuth mapping is available at https://zenodo.org/records/4895404). Below we list details for the datasets:

Naftali Kaminski, Taylor S Adams. [Taylor S Adams, Jonas C Schupp, Sergio Poli, Ehab A Ayaub, Nir Neumark, Farida Ahangari, Sarah G Chu, Benjamin A Raby, Giuseppe Deluliis, Michael Januszyk, Qiaonan Duan, Heather A Arnett, Asim Siddiqui, George R Washko, Robert Homer, Xiting Yan, Ivan O Rosas, Naftali Kaminski] Single-cell RNA-seq reveals ectopic and aberrant lung-resident cell populations in idiopathic pulmonary fibrosis. GEO—GSE136831, https://doi.org/10.1126/sciadv.aba1983 (2020).

Jonathan A Kropski, Arun C Habermann. [Arun C Habermann, Austin J Gutierrez, Linh T Bui, Stephanie L Yahn, Nichelle I Winters, Carla L Calvi, Lance Peter, Mei-I Chung, Chase J Taylor, Christopher Jetter, Latha Raju, Jamie Roberson, Guixiao Ding, Lori Wood, Jennifer M S Sucre, Bradley W Richmond, Ana P Serezani, Wyatt J McDonnell, Simon B Mallal, Matthew J Bacchetta, James E Loyd, Ciara M Shaver, Lorraine B Ware, Ross Bremner, Rajat Walia, Timothy S Blackwell, Nicholas E Banovich, Jonathan A Kropski] Single-cell RNA sequencing reveals profibrotic roles of distinct epithelial and mesenchymal lineages in pulmonary fibrosis. GEO—GSE135893, https://doi.org/10.1126/sciadv.aba1972 (2020).

Alexander V Misharin, Paul A Reyfman. [Paul A Reyfman, James M Walter, Nikita Joshi, Kishore R Anekalla, Alexandra C McQuattie-Pimentel, Stephen Chiu, Ramiro Fernandez, Mahzad Akbarpour, Ching-I Chen, Ziyou Ren, Rohan Verma, Hiam Abdala-Valencia, Kiwon Nam, Monica Chi, SeungHye Han, Francisco J Gonzalez-Gonzalez, Saul Soberanes, Satoshi Watanabe, Kinola J N Williams, Annette S Flozak, Trevor T Nicholson, Vince K Morgan, Deborah R Winter, Monique Hinchcliff, Cara L Hrusch, Robert D Guzy, Catherine A Bonham, Anne I Sperling, Remzi Bag, Robert B Hamanaka, Gökhan M Mutlu, Anjana V Yeldandi, Stacy A Marshall, Ali Shilatifard, Luis A N Amaral, Harris Perlman, Jacob I Sznajder, A Christine Argento, Colin T Gillespie, Jane Dematte, Manu Jain, Benjamin D Singer, Karen M Ridge, Anna P Lam, Ankit Bharat, Sangeeta M Bhorade, Cara J Gottardi, G R Scott Budinger, Alexander V Misharin] Single-Cell Transcriptomic Analysis of Human Lung Provides Insights into the Pathobiology of Pulmonary Fibrosis. https://doi.org/10.1164/rccm.201712-2410OC (2019).

Robert Lafyatis, Christina Morse. [Christina Morse, Tracy Tabib, John Sembrat, Kristina L Buschur, Humberto Trejo Bittar, Eleanor Valenzi, Yale Jiang, Daniel J Kass, Kevin Gibson, Wei Chen, Ana Mora, Panayiotis V Benos, Mauricio Rojas, Robert Lafyatis] Proliferating SPP1/MERTK-expressing macrophages in idiopathic pulmonary fibrosis. GEO—GSE128033, https://doi.org/10.1183/13993003.02441-2018 (2019).

## Declarations

### Ethics approval and consent to participate
Not applicable.

### Consent for publication
Not applicable.

### Competing interests
A.M. has been an employee of Genentech since August 2023. J.C.M. has been an employee of Genentech since September 2022.

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
