## [Additional file 7: Peer review history · Genome Biology]

Review History

First round of review

Reviewer 1

Were you able to assess all statistics in the manuscript, including the appropriateness of statistical tests used? Yes: Statistical testing was appropriately described.

Were you able to directly test the methods? Yes.

Comments to author:

In the manuscript, "Leveraging neighbourhood representations of single-cell data to achieve sensitive DE testing" by Missarova et al. the authors propose a cluster-free statistical framework for differential expression testing across multi-condition single-cell datasets. Previous methods for identifying differentially expressed genes have largely relied on cell type comparisons between conditions (e.g. disease, control); however, cell type annotations are challenging to determine, thus this approach may fail to detect differential gene expression dynamics that occur at a finer or more continuous resolution. Moreover, cell types may be differentially abundant across conditions leading to false positives. To overcome these limitations, the authors present an alternative approach that performs differential expression testing by assigning cells to overlapping cellular neighborhoods on a refined '2nd order' k-nearest neighbor graph. They demonstrate how this approach, in combination with differential abundance testing, can be used to find gene expression modules associated with disease progression. Furthermore they highlight important considerations when performing differential expression vs. differential abundance testing. Overall, this manuscript presents a novel method that is broadly applicable to the single-cell community. However, it currently lacks a rigorous comparison of its performance with respect to alternative strategies. I have a few comments that should be clarified and recommend additional analyses that may improve the interpretability of the approach.

Major comments:

1. On page 4, the authors propose a 2nd order k-nearest neighbor graph representation for preserving rare cell states. Furthermore, in Supplementary Figure 2, they demonstrate how this approach achieves higher cell type purity scores across neighborhoods for rare cell states than a traditional k-nearest neighbor graph. How does this edge augmentation approach impact the purity and subsequent detection of DE genes in abundant, yet less transcriptionally distinct cell states? Given that those cellular neighborhoods are now more densely connected to one another, and there is a tradeoff between neighborhood size and homogeneity, it is recommended that the authors show cell type purity and DE detection power across all cell states using 1st and 2nd order graphs.
2. Although the authors propose a novel method for differential expression testing and highlight use cases for a neighborhood-level analysis, they provide no comparisons to existing strategies on real or simulated data. It is recommended that the authors benchmark their method against alternative strategies for DE detection (e.g. single-cell frameworks - Cacoa: Petukov et al BioRxiv 2022 and clustering-based frameworks) by measuring TPR/FPR/AUC on simulated data where the ground truth differential expression is known. This would further illustrate and supplement their claims in paragraph 2 on Page 3 in the Introduction section. Moreover, in

reference to the Macrophage Activation During Idiopathic Pulmonary Fibrosis subsection in the Results section, the authors highlight how sub-cell type compositional biases might lead to false positive detection of DE genes requiring a cluster-free approach. It would be useful to compare the identification of DE genes from their cluster-free approach to a cluster-based approach.

3. On Page 10 lines 55-59, the authors reference how it is impossible to characterize different continuous patterns of DE changes within cell types. This statement is a bit misleading as there are methods for detecting continuous differential gene expression patterns at the cell-level for multi-sample multi-condition datasets (e.g. Lamian: Hou et. al. Nature Communications 2023). Can Milo-DE be used for detecting continuous differential gene expression changes across case/control samples? This would be an interesting application and comparison.

4. How sensitive is Milo-DE to choice of index cell selection? Are log fold changes and corrected p-values reproducible for cells in similar neighborhoods?

5. Milo-DE depends on a series of preprocessing steps and hyperparameters. Although the authors provide a discussion of embedding choice and neighborhood size in the Supplementary Notes, it would be helpful to provide a subsection or table that consolidates practical recommendations for parameter selection (e.g. class imbalance, min gene expression counts, HVG selection, 1st vs. 2nd order graphs, choice of k in k-nearest neighbor graph construction, choice of embedding, spatial FDR cutoff, etc). This, as well as the simulation/benchmarking studies, should be referenced in the main text. Furthermore, in reference to Supplementary Note 2, Figure 1, the authors provide a simulation study to highlight how the number of replicates or class imbalance may impact the sensitivity or specificity of DE detection. Although the authors reference the optimal k-neighborhood size, they should further comment and clarify in the text how class imbalance (e.g. more control than case samples) should be handled. Moreover, these supplementary figures are a bit blurry and hard to read in scale and explanation.

6. To ensure reproducibility, the authors should provide package versions for all software used, including the version of R and versions of individual functions used in analyses. Similarly, on pages 19, 24, and 26 in the Methods section, the authors should provide precise details regarding the estimated parameters used in all splatter simulation studies.

Minor comments:

1. Supplementary Figure 2 depicts the average neighborhood size for 1st and second-order kNN graphs. How were the order-k values and neighborhood sizes chosen on the x-axes? How are they comparable across 1st and 2nd order graphs?

2. On Page 6-7, the authors perform a series of post hoc filtering criteria to reduce the number of tests for increased statistical power and scalability of analysis. Namely, first cells are subsampled with a neighborhood refinement step to ensure 'complete coverage' of the graph. Then genes with low expression across conditions are removed. Lastly, 'uninteresting' neighborhoods are discarded. Rather than sampling cells according to the graph and then further removing redundant neighborhoods, have the authors considered performing representative downsampling or single-cell sketching prior to graph construction? For example, previous work has focused on identifying subsets of cells that preserve the original distribution of cell states (e.g. Kernel

Herding sketching: Baskaran et al ACM BCB 2022) or preserving transcriptional diversity by minimizing the Hausdorff distance between sets (e.g. Hopper: DeMeo et al Bioinformatics 2020, Geometric Sketching: Hie et al Cell Systems 2019).

3. In reference to the Neighborhood Assignment and Neighborhood Refinement subsections in the Methods section, it is recommended that the authors further clarify how cells are assigned to neighborhoods. Are neighborhoods represented by the index cell, such that cells that are connected by an edge to the index cell in the original graph are considered to be included in that neighborhood? In this way, cells can belong to multiple neighborhoods? What does it mean for a cell to be unassigned to a neighborhood? Is a new kNN graph constructed amongst index cells for DE testing?

4. On page 10 line 8, we also observe 'sign' differences -> 'significant' differences

5. The authors reference on several occasions in the methods a 'p-value' (e.g. page 25 lines 30 - p-value < 0.01, page 24 line 2 - p-value < 0.05, page 35 line 45 - p-value < 0.05, page 28 line 58 - p-value < 0.1). It should be clarified if this p-value is the neighborhood and gene-wise corrected p-value. How are these significance thresholds chosen? Why are they different?

6. All abbreviations should be formally defined (e.g. MNN - mutual nearest neighbors on page 21 line 56, PCA - principal components analysis on page 20 line 57, kNN - k-nearest neighbor graph).

Reviewer 2

Were you able to assess all statistics in the manuscript, including the appropriateness of statistical tests used? Yes: All analysis details are well-described.

Were you able to directly test the methods? Yes.

Comments to author:

The manuscript "Leveraging neighbourhood representations of single-cell data to achieve sensitive DE testing" proposes the method miloDE to address issues in cell-type specific differential expression testing in scRNA-seq data. Rather than relying on only imperfect cell-type annotations for DE testing, miloDE constructs a cell graph and performs testing within neighbourhoods.

1. I think the method is potentially quite useful and informative. However, the first few paragraphs in the Results section are difficult to get through. Adjusting these would greatly improve understanding and flow of the manuscript. In most of the cases below, stating the purpose and then the details (instead of the other way around) would likely be a sufficient solution or consider moving some details to the Methods section.

a. The explanation of the method in the first Results paragraph is a bit too superficial given the subsequent paragraphs. Specifically, if DE is tested per neighbourhood, how exactly are the overlapping neighbourhoods being leveraged? The sentences do not connect well to assist understanding.

b. In contrast, the next two paragraphs containing supplemental notes have too much detail and without understanding the method better initially, they are too dense. These seem more appropriate in the Methods section or at least some details could be moved to Methods.

For example, the sentence "As k increases, the average neighbourhood size for the 2nd-order kNN graph increases considerably faster than it does for the 1st-order, with a neighbourhood size in the low hundreds being achieved when k is between 20 and 30"

-What does this mean to the reader? Is this good for the proposed method?

c. Why is it important to restrict the average neighbourhood size to the low hundreds cell target? (This is noted in a paragraph in the Results section on the chimeric mouse embryo dataset.)

d. The assignment is noted as being done 3 times. Is this just for the simulation or for any analysis?

e. Finally, at the end of these paragraphs, it became clear that the purpose was to show that the neighbourhoods generally have high cell type purity using the 2nd-order KNN approach. Stating this initially would be more helpful. Although, it is not clear why this was done only for rare cell types?

f. In the paragraph beginning "Finally, we examined how cell type purity affects the sensitivity of DE detection", there are no sentences explaining how DE is actually performed using miloDE prior to this or in the paragraph.

2. The synthetically perturbed/simulated data finds that even using a purity threshold of .1 leads to sufficient testing power. But as noted this also depends greatly on the effect size. Instead of just saying "high sensitivity", it would be helpful to add the average sensitivity to the text for a given effect size, i.e. average 80% sensitivity with $\log_2\text{FC} > 2$.

3. Any explanation for why the number of cells appears to have a larger effect than the number of biological replicates regarding DE sensitivity and specificity in Supplemental Note 2? This seems a bit surprising since miloDE creates pseudo bulks for each biological replicate when testing. (It could just be the lower figure quality in the supplement submission, the lines are a bit hard to decipher.)

Also, the Wang 2019 reference treats each cell as a 'replicate' which miloDE does not, so it is not totally clear why the replicates number does not have a larger effect here.

Minor:

1. In Methods section 2.2, line 30 should read "no gene selection will [be] performed"

2. Is the "bulk DE" approach mentioned in the caption to Supplementary Figure 6, actually meant to say "pseudo-bulk DE"?

We thank each of the reviewers for their insightful comments on our work. Please find below specific responses and actions taken to address their concerns.

Black Arial corresponds to the original reviewer's comment.

Blue Arial corresponds to our response

Red italics correspond to specific changes made to the manuscript (alternatively, we refer to the section of the manuscript that has been substantially rewritten)

Reviewer #1:

In the manuscript, "Leveraging neighbourhood representations of single-cell data to achieve sensitive DE testing" by Missarova et al. the authors propose a cluster-free statistical framework for differential expression testing across multi-condition single-cell datasets. Previous methods for identifying differentially expressed genes have largely relied on cell type comparisons between conditions (e.g. disease, control); however, cell type annotations are challenging to determine, thus this approach may fail to detect differential gene expression dynamics that occur at a finer or more continuous resolution. Moreover, cell types may be differentially abundant across conditions leading to false positives. To overcome these limitations, the authors present an alternative approach that performs differential expression testing by assigning cells to overlapping cellular neighborhoods on a refined '2nd order' k-nearest neighbor graph. They demonstrate how this approach, in combination with differential abundance testing, can be used to find gene expression modules associated with disease progression. Furthermore they highlight important considerations when performing differential expression vs. differential abundance testing. Overall, this manuscript presents a novel method that is broadly applicable to the single-cell community. However, it currently lacks a rigorous comparison of its performance with respect to alternative strategies. I have a few comments that should be clarified and recommend additional analyses that may improve the interpretability of the approach. However, it currently lacks a rigorous comparison of its performance with respect to alternative strategies. I have a few comments that should be clarified and recommend additional analyses that may improve the interpretability of the approach.

We thank the reviewer for their appreciation of the potential utility of our approach.

Major comments:

1. On page 4, the authors propose a 2nd order k-nearest neighbor graph representation for preserving rare cell states. Furthermore, in Supplementary Figure 2, they demonstrate how this approach achieves higher cell type purity scores across neighborhoods for rare cell states than a traditional k-nearest neighbor graph. How does this edge augmentation approach impact the purity and subsequent detection of DE genes in abundant, yet less transcriptionally distinct cell states? Given that those cellular neighborhoods are now more densely connected to one another, and there is a tradeoff between neighborhood size and homogeneity, it is recommended that the authors show cell type purity and DE detection power across all cell states using 1st and 2nd order graphs.

We apologise for this oversight. The focus, in our original submission, on rare cell types was motivated by the intuition that these cell types will be the ones that are primarily impacted by mixing different cell types. However, we agree that the issue might also propagate more broadly and affect even abundant cell types if they extensively mix with other, transcriptionally similar cell types. Therefore, building upon the reviewer's suggestion, we extended and improved the original analysis in the following ways:

- a) We extended the simulations to better represent multiple different cell types, ranging from highly rare to abundant.
- b) We added a complementary metric to estimate the overall homogeneity of a given neighbourhood assignment. While the metric we focused on in our original submission – the cell type purity score – provides insight into whether any individual cell type consistently mixes with other cell types for a given neighbourhood assignment, it does not reflect the overall homogeneity across all the neighbourhoods in the assignment. To account for this, we added a second metric – relative cell type enrichment. Specifically, we first identify, for each neighbourhood, its most abundant cell type and then report the percentage of cells in the neighbourhood annotated with this cell type. We reasoned that more 'homogenous assignments' are likely to be represented by neighbourhoods with higher relative cell type enrichment (this can be compared across all neighbourhoods and per individual cell types).

The overall conclusions of the extended analysis are that:

- a) The distributions of cell type purity and the number of cells per neighbourhood (two factors affecting the ability to detect DE genes) are highly similar for abundant cell types between 1st and 2nd order assignments, resulting in very similar DE detection (new Supp. Fig. 2D, new Supp. Fig. 3). As previously, it differs for rare cell types, in favour of 2nd order assignments.

b) The relative cell type enrichment is highly similar between 1st and 2nd order for abundant cell types and is slightly higher for rare cell types in 2nd order assignment (new Supp. Fig. 2C).

Consequently, we conclude that 2nd order assignment does not compromise the homogeneity of neighbourhoods for abundant cell types, and that it increases the power to detect DE genes for rare cell types. Specifically, for the rarest cell type - Primordial germ cells - we achieve a maximum detection of 0.78 in the 2nd-order kNN against a maximum detection of 0.63 in the 1st-order kNN. The results of this analysis are described in the updated 1st Section (Supp. Fig. 2,3).

2. Although the authors propose a novel method for differential expression testing and highlight use cases for a neighborhood-level analysis, they provide no comparisons to existing strategies on real or simulated data. It is recommended that the authors benchmark their method against alternative strategies for DE detection (e.g. single-cell frameworks - Cacao: Petukov et al BioRxiv 2022 and clustering-based frameworks) by measuring TPR/FPR/AUC on simulated data where the ground truth differential expression is known. This would further illustrate and supplement their claims in paragraph 2 on Page 3 in the Introduction section.

We thank the reviewer for this important suggestion. We have extended the analyses of the performance of miloDE on simulated data (2nd Section) and added a comparison with Cacao and the pseudo-bulk approach using simulations with the known ground truth.

Since the outputs of Cacao and miloDE are not immediately comparable (single-cell level against neighbourhood level), we had to process the outputs to achieve comparable entities. In brief, we performed the comparison on the neighbourhood- and single cell-levels. For the neighbourhood comparison, we used the immediate outputs of miloDE; for Cacao, for each neighbourhood, we calculated average z-scores across all cells from the neighbourhood. For the single-cell comparison, we used the immediate outputs from Cacao, and for miloDE we 'decomposed' the outputs from miloDE to return the statistic for each cell. Specifically, we first transformed corrected (across neighbourhood) p-values into z-scores, and then, for each cell, we calculated the average z-score across all neighbourhoods to which it was assigned. The main conclusion is that Cacao, when compared to miloDE, is more impacted by multiple testing correction as well as being more susceptible to single-cell noise. Specifically, if a fraction of perturbed cells is less than 5%, adjusting z-scores in Cacao eliminates any significance (AUC = 0.5) while AUCs in miloDE remain high even for the

datasets with the smallest perturbed fraction (0.5%). We also note that when performing a comparison at the single-cell level, even for unadjusted z-scores from Cacoa, AUCs are higher in miloDE (average AUC = 0.93) compared to Cacoa (average AUC = 0.72). We hypothesize that this could be explained by the sensitivity of Cacoa to the inherent noise in single-cell data.

Similarly, the outputs of the pseudo-bulk approach and miloDE are not directly comparable. To attempt to address this, we estimated the sensitivity of the pseudo-bulk approach as a binary value (significant/not significant), whereas sensitivity for miloDE was estimated as a continuous value between 0 and 1 across the neighbourhoods. Overall, as expected, the sensitivity of the pseudo-bulk approach depends, to a greater extent than miloDE, on the real fraction of DE genes. Specifically, when the fraction of perturbed cells is small (below 5%), even a high effect size can not be detected by the pseudo-bulk approach.

The corresponding analyses are described in the updated Section 2 and Supplementary Figures 5, 6, 7, 8.

Moreover, in reference to the Macrophage Activation During Idiopathic Pulmonary Fibrosis subsection in the Results section, the authors highlight how sub-cell type compositional biases might lead to false positive detection of DE genes requiring a cluster-free approach. It would be useful to compare the identification of DE genes from their cluster-free approach to a cluster-based approach

We apologise for the initial lack of sufficient comparative analyses between the pseudo-bulk and miloDE approaches. To address this, we have now extended this comparison in the IPF context, focusing particularly on genes that are identified as significantly and strongly DE at the pseudo-bulk level ($FDR < 0.1$), but which are not significant in any miloDE neighbourhood (corrected across neighbourhoods p -value < 0.1 ; see the added paragraph in Section 4 for more detail).

Our analyses suggest that there are two (not mutually exclusive) reasons that could drive such results. On one hand, the likely FPs in the pseudo-bulk analyses that are driven by sporadic high expression in a limited number of cells appear to be, at least partially, not detected in miloDE. On the other hand, as discussed in the manuscript, the sensitivity of miloDE is limited by the number of cells per neighbourhood, and this can be (at least partially) rescued by aggregating more cells in the pseudo-bulk approach. Importantly, although both approaches, naturally, have some limitations we want to emphasise that

miloDE enables in-depth post hoc analysis (e.g. gene clustering) that is much more challenging using pseudo-bulk approaches.

Additionally, we examined whether sub-cell type compositional biases might lead to the false positive detection of DE genes in the pseudo-bulk approach (the issue originally addressed by the reviewer). To do so, we identified two sub-cell states based on Spp1-expression, since Spp1+ has been shown to be predictive of the differential abundance (DA). Next, for each gene we estimated logFC between Spp1+ and Spp1- cells while controlling for the disease status. We refer to this metric as 'DA marking potential'. We next focused our analyses on genes that are detected as significant in the pseudo-bulk approach, but have no significantly DE neighbourhoods in miloDE. We do not observe that these genes are positively biased for 'DA marking potential'. Therefore, we suggest that at least in this particular setup (disease/cell type/dataset), DA and sub-cell type differences in composition are not a driving force for discrepancies between pseudo-bulk and miloDE. More broadly, we want to emphasise that it is possible that a gene both marks a DA sub-region(s) and is simultaneously DE between control and case conditions, either in one or several DA sub-regions. Moreover, in a disease setting, it is likely that genes that drive the disease state, will both mark DA sub-regions (e.g. disease-state sub-region and healthy-state sub-region) and DE between disease and healthy cells.

Overall, we thank the reviewer for suggesting this analysis since it made us revise the importance of sub-cell type compositional biases for false positive DE detection. We now updated the corresponding statements in the manuscript. Specifically, the updated text goes as follows:

On a molecular level, one axis of macrophage variability in IPF is associated with the expression of the fibrotic marker Spp1 (Morse et al. 2019; Reyfman et al. 2019; Adams et al. 2020). The abundance of an Spp1-high subpopulation has been repeatedly observed in IPF patients, however, Spp1-high macrophages also exist in healthy patients (albeit at a lower abundance and absolute Spp1 expression). We propose that miloDE is a suitable tool to study complex DE patterns with respect to various axes of disease-driven variability such as DA (by paired Milo - miloDE analyses) or Spp1 expression. Additionally, while using miloDE output, we can employ clustering algorithms to characterise various 'co-regulated' DE patterns arising in this population of cells during disease progression.

3. On Page 10 lines 55-59, the authors reference how it is impossible to characterize different continuous patterns of DE changes within cell types. This statement is a bit

misleading as there are methods for detecting continuous differential gene expression patterns at the cell-level for multi-sample multi-condition datasets (e.g. Lamian: Hou et. al. Nature Communications 2023). Can Milo-DE be used for detecting continuous differential gene expression changes across case/control samples? This would be an interesting application and comparison.

We agree with the reviewer and apologise for the misleading wording here. While we claim that pseudo-bulk DE methods don't provide the resolution to cluster genes and study patterns of co-regulation, it is misleading to say that it is impossible to characterise continuous patterns of DE changes with other approaches, such as differential pseudotime analysis as suggested by the reviewer. To improve the clarity of our manuscript, we have updated the text accordingly (see below).

More generally, we do not consider miloDE, in its current form, to be immediately applicable to detecting continuous differential gene expression changes. Such analysis could be performed post hoc, by for example pairing miloDE outputs with pseudotime trajectory inference and studying the changes in neighbourhoods along the identified trajectories. In general, we suggest that miloDE provides a hypothesis-free approach to learn DE expression vectors in a local, heterogeneity-controlled manner, and it provides an output that can be further interpreted and explored based on the underlying hypothesis for the data in hand. With this, miloDE is not directly comparable with Lamian, and therefore we suggest that the benchmarking these two methods lies outside the scope of our study. However, we agree that differential pseudotime trajectory analysis is a highly relevant approach to studying disease-induced cell state changes and we now have incorporated the reference to Lamian in the discussion section:

While miloDE approximates gene regulation as a more continuous function of the manifold, it is important to acknowledge that it leverages the same principle of cell grouping as standard per cell type approaches. Therefore, in cases where a manifold is comprised of fairly distinctive, homogenous cell types, high-resolution clustering coupled with a per bulk DE approach will likely yield very similar results to miloDE. Similarly, in the case of continuous trajectories, a per bulk approach combined with the inclusion of a continuous covariate representing a pseudotime (or, perhaps, a more complex and bespoke approach such as tradeSeq (Van den Berge et al. 2020)) might also yield similar results and resolution to miloDE. Moreover, differential pseudotime analysis such as Lamian and others can provide the insights for how certain lineages are affected, both in abundance and expression, during the disease or perturbation (Hou et al. 2023; Campbell and Yau 2018). While powerful in

some cases, the trajectory-based branch of methods assumes that the manifold is continuous. In reality, a single-cell manifold is typically comprised of an intricate and complex combination of discrete and continuous cell types, and, moreover, it is often unclear whether the annotated cell types contain further heterogeneity. Combined, we suggest that miloDE allows the user to overcome laborious and bespoke analysis for each individual cell type using an elegant and straightforward approach.

4. How sensitive is Milo-DE to choice of index cell selection? Are log fold changes and corrected p-values reproducible for cells in similar neighborhoods?

We thank the reviewer for bringing up this important point. Since the neighbourhood assignment is semi-random (i.e. it relies on a random selection of cells that is then further refined to select more densely connected cells as neighbourhood centres), it is important to test for the robustness of the final DE detection with respect to this assignment. We note that we provide such testing at each aspect of the method benchmark. Specifically,

a) In the analyses of the importance of the embedding, for each parameter choice, we perform an assignment several times and show that the type of embedding has a more pronounced effect than the neighbourhood assignment.

b) In the analyses of the importance of the order choice, we performed several assignments for each parameter set, and we showed higher cell type purity (and therefore power to detect DE) for the 2nd order neighborhood approach for rare cell types across all replicates of neighbourhood assignments.

c) Analogously, multiple choices of parameters for neighbourhood construction were assessed when using the simulated datasets to compare MiloDE against Cacoa and the pseudo-bulk approaches. While the stochasticity in neighbourhood assignment has a measurable effect in edge cases with a very limited number of ground truth DE neighbourhoods, overall we see robust patterns across the tested conditions. Importantly, we conclude that the neighbourhoods that have a similar fraction of 'perturbed' cells are likely to yield similar DE detection.

5. Milo-DE depends on a series of preprocessing steps and hyperparameters. Although the authors provide a discussion of embedding choice and neighborhood size in the Supplementary Notes, it would be helpful to provide a subsection or table that consolidates practical recommendations for parameter selection (e.g. class imbalance, min gene expression counts, HVG selection, 1st vs. 2nd order graphs, choice of k in k-nearest neighbor graph construction, choice of embedding, spatial FDR cutoff, etc). This, as well as

the simulation/benchmarking studies, should be referenced in the main text. Furthermore, in reference to Supplementary Note 2, Figure 1, the authors provide a simulation study to highlight how the number of replicates or class imbalance may impact the sensitivity or specificity of DE detection. Although the authors reference the optimal k-neighborhood size, they should further comment and clarify in the text how class imbalance (e.g. more control than case samples) should be handled. Moreover, these supplementary figures are a bit blurry and hard to read in scale and explanation.

We thank the reviewer for the suggestion as we strive to make our method easy to implement in a useful and sensible way. As pointed out, we comment on various aspects of the parameter choice throughout the manuscript, and we agree that it will be useful to combine all our recommendations in a single place, that is easily accessed. To achieve this, we have now created Supplementary Figure 14, which highlights the individual steps of the miloDE algorithm (covering graph representation, neighbourhood assignment, and DE testing itself) and the parameters that affect each step. We elaborated on our reasoning for the importance of each parameter and provided recommendations on how to proceed in each step.

Particularly, we want to highlight that perhaps the most important parameter is a choice of k since it controls the average neighbourhood size, which in turn affects the sensitivity of DE detection. Importantly, the optimal neighbourhood size is highly dependent on the parameters of the dataset such as overall dataset variance, number of samples, and, as pointed out by the reviewer, class imbalance, as well as the effect size that is aimed to be detected. Accordingly, the optimal choice of k becomes a two-step process:

- a) Evaluating which average neighbourhood size will allow the desired level of sensitivity and specificity for a given dataset. To assist with this, we now recorded the estimated DE detection from the simulations performed in the analysis for Supplementary Note 2 (Supp. Table 1). We also reference Supp. Table 1 in Supp. Figure 14.
- b) Evaluating the relationship between different values of k and the neighbourhood size distribution. To provide means for this evaluation, we provide a helper function `miloDE::estimate_neighbourhood_sizes.R` that allows a user to estimate neighbourhood size distribution as a function of k.

Consequently, we reference Supp. Fig. 14 and Supp. Table 1 in the Section 1, Supplementary Note 2 and Methods:

Above we described each step of the pipeline. To facilitate the appropriate use of the algorithm, we combined the recommendations for the optimal selection of the hyperparameters of the algorithm (Supp. Fig. 14 and Supp. Table 1). We suggest that the most appropriate parameters to consider are the different order-k combination which define the distribution of neighbourhood sizes and thus impact the sensitivity of DE detection. Supp. Table 1 provides a reference table that allows DE to be estimated as a function of effect size, number of tested cells and replicates. Additionally, to estimate how neighbourhood size distribution depends on order-k for any specific dataset, a corresponding function `miloDE::estimate_neighbourhood_sizes` is available within the R package.

As for the class imbalance, we agree that this is an important aspect to consider while estimating the appropriate neighbourhood size and k. We have extended the original simulations and performed a more controlled comparison for class imbalance (updated Supplementary note 2). The main conclusion is that when controlling for the total number of cells, higher class imbalance (as expected) results in lower sensitivity. We, therefore, suggest that the total number of cells combined with the minimum number of cells (across the conditions) are two bottlenecks that are important to consider while performing miloDE and choosing appropriate k.

6. To ensure reproducibility, the authors should provide package versions for all software used, including the version of R and versions of individual functions used in analyses. Similarly, on pages 19, 24, and 26 in the Methods section, the authors should provide precise details regarding the estimated parameters used in all splatter simulation studies.

All of the scripts used are available in the public GitHub folder: https://github.com/MarioniLab/miloDE_analysis (also referenced in the manuscript). Additionally, this folder contains the subfolder `session_info` which in turn contains an HTML document `session_info_R.html` that references the versions of all R packages while `session_info_python.txt` references the versions of all Python packages. Details of the simulations are described in the Method section, and specific splatter-estimated parameters of simulations are listed in Supplementary Table 4.

Minor comments.

1. Supplementary Figure 2 depicts the average neighborhood size for 1st and second-order kNN graphs. How were the order-k values and neighborhood sizes chosen on the x-axes? How are they comparable across 1st and 2nd order graphs?

The rationale for the choice of [order-k] stems from the analysis in Supplementary Note 2. In this analysis, we first identify that a few hundred cells is a reasonable target for the neighbourhood size in order to ensure high sensitivity. Accordingly, we select the values for [order-k] such that they return the average neighbourhood size of the selected range of few hundred cells. The feature we are comparing between the 1st- and 2nd-order settings is the number of cells per neighbourhood.

2. On Page 6-7, the authors perform a series of post hoc filtering criteria to reduce the number of tests for increased statistical power and scalability of analysis. Namely, first cells are subsampled with a neighborhood refinement step to ensure 'complete coverage' of the graph. Then genes with low expression across conditions are removed. Lastly, 'uninteresting' neighborhoods are discarded. Rather than sampling cells according to the graph and then further removing redundant neighborhoods, have the authors considered performing representative downsampling or single-cell sketching prior to graph construction? For example, previous work has focused on identifying subsets of cells that preserve the original distribution of cell states (e.g. Kernel Herding sketching: Baskaran et al ACM BCB 2022) or preserving transcriptional diversity by minimizing the Hausdorff distance between sets (e.g. Hopper: DeMeo et al Bioinformatics 2020, Geometric Sketching: Hie et al Cell Systems 2019).

We agree with the reviewer that there are several ways to approach representative sampling of cells from the population. As pointed out by the reviewer, the preservation of rare cell states is important, and in the context of miloDE, we particularly want to ensure the homogeneity of neighbourhoods that capture rare cell states. We suggest that the approach we take - sampling an excessive number of index cells to ensure that all cells are assigned followed by the neighbourhood filtering and combined with the implementation of 2nd-order kNN for better recapitulation of the local density achieves this goal. As highlighted by the reviewer, geometric sketching could have yielded a similar outcome. However, since geometric sketching is likely to greatly shift the composition of the cell types/states, with potential to impact the significance of estimates after multiple testing correction, we feel that such a comparison is beyond the scope of the current manuscript. Moreover, and importantly, as shown by our simulations, the approach we take strikes a good balance between computational complexity and sensitivity to detect DE genes.

3. In reference to the Neighborhood Assignment and Neighborhood Refinement subsections in the Methods section, it is recommended that the authors further clarify how cells are assigned to neighborhoods. Are neighborhoods represented by the index cell, such that call

ells that are connected by an edge to the index cell in the original graph are considered to be included in that neighborhood? In this way, cells can belong to multiple neighborhoods? What does it mean for a cell to be unassigned to a neighborhood? Is a new kNN graph constructed amongst index cells for DE testing?

We thank the reviewer for this suggestion. To address this, we have extended the relevant Method section accordingly. Specifically, the graph (both 1st and 2nd) is constructed using all cells, and the index cells are used to select neighbourhoods (one neighbourhood corresponds to one index cell and consists of the index cell and all its neighbours in the final graph). Cells can indeed belong to multiple neighbourhoods and we refer to this property as an overlap between the neighbourhoods. Accordingly, it is also possible that a cell does not belong to any neighbourhood and we refer to these cells as unassigned cells.

4. On page 10 line 8, we also observe 'sign' differences -> 'significant' differences

We apologise for the confusion caused by bad wording here. What we intended to state is that neighbourhoods in this region have both positive and negative effect sizes (hence sign difference), and overall DA is strongly correlated with the transcriptional proximity to the blood progenitors. We have now updated the original text to achieve better clarity:

In agreement with this, we observe a strong correlation between the distance to blood progenitors and enrichment and DA. Specifically, neighbourhoods that are most proximal to blood progenitors neighbourhoods are significantly enriched for Tal1+ cells, which in turn are 'preceded' by the neighbourhoods that are significantly enriched for Tal1- cells.

5. The authors reference on several occasions in the methods a 'p-value' (e.g. page 25 lines 30 - p-value < 0.01, page 24 line 2 - p-value < 0.05, page 35 line 45 - p-value < 0.05, page 28 line 58 - p-value < 0.1). It should be clarified if this p-value is the neighborhood and gene-wise corrected p-value. How are these significance thresholds chosen? Why are they different?

We thank the reviewer for pointing this out. We now ensure that we use only one, generally accepted, significance threshold for raw p-values (0.05) and we use one significance threshold for corrected, either across neighbourhoods or genes, p-values (0.1). The choice of whether to use corrected p-values (either genes or neighbourhoods) or raw p-values was driven by the specific hypothesis/question that is addressed in each analysis. Please note that in the original IPF analyses, we used a threshold of 0.05 for the corrected p-value. For

consistency, we have now adjusted this to 0.1, and to prioritise gene candidates for further investigation, we have increased the cut-off for the fraction of significantly DE neighbourhoods to 0.25. Please see the updated IPF analyses in Section 4.

6. All abbreviations should be formally defined (e.g. MNN - mutual nearest neighbors on page 21 line 56, PCA - principal components analysis on page 20 line 57, kNN - k-nearest neighbor graph).

We now ensure that all abbreviations are defined prior to the usage. Specifically,

- Differential expression - DE - is defined in the abstract (page 1)
- single-cell RNA sequencing - scRNA-seq - is defined in the abstract (page 1)
- Differential abundance - DA - is defined in the Introduction (page 2)
- k-Nearest Neighbours - kNN - is defined in the Introduction (page 3)
- log Fold Change - logFC - is defined in Section 1 (page 4)
- Mutual nearest neighbours - MNN - is defined in Section 1 (page 5)
- Primordial Germ cells - PGCs - is defined in Section 1 (page 7)
- False Discovery Rate - FDR - is defined in Section 2 (page 8)
- Idiopathic Pulmonary Fibrosis - IPF - is defined in Section 4 (page 11)
- Principal component analysis - PCA - is defined in the Methods (page 21)
- Area under the curve - AUC - is defined in Section 2 (page 10)

Reviewer #2: The manuscript "Leveraging neighbourhood representations of single-cell data to achieve sensitive DE testing" proposes the method miloDE to address issues in cell-type specific differential expression testing in scRNA-seq data. Rather than relying on only imperfect cell-type annotations for DE testing, miloDE constructs a cell graph and performs testing within neighbourhoods.

1. I think the method is potentially quite useful and informative. However, the first few paragraphs in the Results section are difficult to get through. Adjusting these would greatly improve understanding and flow of the manuscript. In most of the cases below, stating the purpose and then the details (instead of the other way around) would likely be a sufficient solution or consider moving some details to the Methods section.

a. The explanation of the method in the first Results paragraph is a bit too superficial given the subsequent paragraphs. Specifically, if DE is tested per neighbourhood, how exactly are the overlapping neighbourhoods being leveraged? The sentences do not connect well to assist understanding.

b. In contrast, the next two paragraphs containing supplemental notes have too much detail and without understanding the method better initially, they are too dense. These seem more appropriate in the Methods section or at least some details could be moved to Methods.

For example, the sentence "As k increases, the average neighbourhood size for the 2nd-order kNN graph increases considerably faster than it does for the 1st-order, with a neighbourhood size in the low hundreds being achieved when k is between 20 and 30"

-What does this mean to the reader? Is this good for the proposed method?

c. Why is it important to restrict the average neighbourhood size to the low hundreds cell target? (This is noted in a paragraph in the Results section on the chimeric mouse embryo dataset.)

d. The assignment is noted as being done 3 times. Is this just for the simulation or for any analysis?

e. Finally, at the end of these paragraphs, it became clear that the purpose was to show that the neighbourhoods generally have high cell type purity using the 2nd-order KNN approach. Stating this initially would be more helpful. Although, it is not clear why this was done only for rare cell types?

f. In the paragraph beginning "Finally, we examined how cell type purity affects the sensitivity of DE detection", there are no sentences explaining how DE is actually performed using miloDE prior to this or in the paragraph.

We thank the reviewer for their constructive and helpful suggestions, all of which have helped to make our manuscript as clear as possible. Specifically:

a) We have extended the first paragraph to give a complete overview of the method. This will allow the reader to understand the main steps of the method and thus facilitate the understanding of the following, more fine-grained paragraphs. We have also added a more in-depth explanation of what neighbourhood assignment means from a biological perspective and how it differs from the metacell representation.

b) We further moved some of the more methodological aspects considering DE testing itself to the methods section. As for some of the aspects considering neighbourhood assignment, on balance we feel that it is still best to keep them in the first section of the Results. Our rationale for this is that neighbourhood assignment dictates how cells will be grouped, which in turn defines the sensitivity of DE detection. Understanding how one relates to the other (both a theoretical understanding and the supporting analysis) is critical for the proper implementation of the method. By keeping this analysis in the first section of the Results we can clearly emphasise the importance of some parameters. However, to ensure better readability, we now split the discussed aspects into subsections, thus facilitating the flow of the paper.

c) We agree with the reviewer that we do not initially provide sufficient intuition as to why we strive to restrict neighbourhood sizes until later in the manuscript. Since providing this intuition will facilitate a better understanding of the method and subsequent analysis, we now incorporate it briefly at the beginning of the relevant subsection. Specifically, we add the following statement:

By design, miloDE aims to detect DE on a local level, within individual transcriptional states. Having more cells within the neighbourhoods increases the probability of generating more heterogeneous neighbourhoods, thus hindering the ability to capture distinct transcriptional states. Accordingly, ideal neighbourhood assignments should contain neighbourhoods of the smallest possible size. On the other hand, the power to detect DE using pseudo-bulk approaches such as edgeR is highly dependent on and scales with the number of tested cells (Crowell et al. 2020).

d) As suggested by the reviewer, we now state upfront our motivation for introducing 2nd-order kNN graphs, with further elaboration of the details:

We suggest that the 2nd-order kNN representation returns more homogenous neighbourhoods than the 1st-order kNN representation, while controlling for average neighbourhood size.

Also, as suggested, we have extended the order analysis across multiple cell types. Since the Reviewer #1 raised similar concerns, we addressed this in more detail above (first comment from the Reviewer #1, pages 2-3).

e) Regarding comment d., a single run of miloDE generates one assignment. Accordingly, we perform multiple assignments for some analysis to assess the robustness of the algorithm. Note, that for analysis of real-life data (Tal1- mouse chimeras and macrophages in IPF), the assignment is performed only once. We now more clearly state that the assignments are random and can change between different runs of miloDE:

Additionally, a certain degree of stochasticity, owing to the random selection of index cells, invariably impacts into any neighbourhood assignment, potentially propagating into discrepancies in the DE detection. To address this, in the following simulations, for each [order-k] combination (hereafter referred to as an assignment) we repeated the assignment 10 times.

f) We also updated Supplementary Note 2 to accommodate the reviewer's comment b. and set a specific neighbourhood size target driven by quantitative estimation of the sensitivity for a certain effect size.

2. The synthetically perturbed/simulated data finds that even using a purity threshold of .1 leads to sufficient testing power. But as noted this also depends greatly on the effect size. Instead of just saying "high sensitivity", it would be helpful to add the average sensitivity to the text for a given effect size, i.e. average 80% sensitivity with $\log_2FC > 2$.

We completely agree with the reviewer that quantitative statements are more helpful and more appropriate. We have now updated the manuscript accordingly and provide the quantitative estimate to previously qualitative statements (i.e. high sensitivity).

3. Any explanation for why the number of cells appears to have a larger effect than the number of biological replicates regarding DE sensitivity and specificity in Supplemental Note 2? This seems a bit surprising since miloDE creates pseudo bulks for each biological replicate when testing. (It could just be the lower figure quality in the supplement submission, the lines are a bit hard to decipher.) Also, the Wang 2019 reference treats each cell as a 'replicate' which miloDE does not, so it is not totally clear why the replicates number does not have a larger effect here.

We thank the reviewer for bringing up this interesting point. DE detection is affected by several factors, including the number of cells per replicate (and accordingly total number of cells) as well number of replicates in both conditions and class imbalance. Overall, a very low number of cells per replicate is suboptimal, since the resulting pseudo-bulk counts are less likely to resemble bulk RNA-seq, and thus follow a negative binomial (NB) distribution, which in turn will result in a suboptimal estimation of the parameters of this distribution and DE inference (albeit the degree of uncertainty will depend on the mean and variance of gene expression). On the other hand, a higher number of replicates will enable higher statistical power to detect DE genes. Accordingly, we reason that DE detection should improve with both a higher number of cells per replicate and a higher number of replicates. With this, we want to highlight that in the original version of the manuscript, the x-axis corresponds to the total number of cells across all replicates. Accordingly, while comparing two datasets with different numbers of replicates, we simultaneously vary two parameters: the number of replicates and the average number of cells per replicate (which, by design, will be lower for datasets with a higher number of replicates). Therefore, we extended the original analysis to better compare and assess how DE detection depends on the average number of cells per

replicate and the number of replicates (Supplementary Note 2, Fig. 1A, B, second and fourth columns). In this comparison, we see that both the higher average number of cells per replicate and higher number of replicates have a pronounced effect on the sensitivity of DE detection.

We also want to elaborate further on the original question regarding the comparison between the importance of the total number of tested cells and the number of replicates and provide two potential explanations for the observed dominance of the former. DE detection between two conditions relies on the absolute difference between the means of the replicates from the two conditions, as well as the variances across replicates within each condition. A smaller variance across replicates in one or both conditions provides a higher chance of significant DE detection. Since we employ a pseudo-bulk approach, the total variance for one condition is a function of variance across cells in each replicate and variance across different replicates from the condition.

More formally, for a single condition, let's denote σ_1^2 - theoretical variance between cells within each replicate (for simplicity we assume it is constant across replicates) and σ_2^2 - theoretical variance across replicates. We also denote N_{sample} as the number of samples and N_{cell} as the number of cells per sample. Following the law of total variance, the variance of the overall mean across pseudo-bulks (where, for simplicity, we assume that the pseudo-bulk profile is the average of the cell-level profiles) can be decomposed into two terms: the variance of the expectation of means across replicates and the expectation of variances across replicates. Following the formula for variance of sample means, the former term is equal to σ_2^2/N_{sample} . Since variance across replicates is variance across sample means for each replicate across cells, the latter term can be further derived as $\sigma_1^2/(N_{sample} * N_{cell})$.

Accordingly, the total variance is composed of these two terms: $\sigma_2^2/N_{sample} + \sigma_1^2/(N_{sample} * N_{cell})$, and we note, that $N_{sample} * N_{cell}$ represents the total number of cells in the condition. The contribution of each term to the final variance depends on the difference between σ_1^2 and σ_2^2 . The expectation for the real-life data (i.e. where replicates correspond to different donors) is that $\sigma_2^2 \gg \sigma_1^2$, and, therefore, the number of replicates/donors will indeed be a dominant factor. However, in our original synthetic simulations, we set up replicates with no batch effect which potentially resulted in $\sigma_1^2 \gg \sigma_2^2$, in turn making the

total number of cells a dominant factor (with the inverse relationship with the variance). We now extended our simulations to introduce batch effect between replicates (Supplementary Note 2, Fig. 1C), and we observe that in the presence of batch effect (bottom row) the difference in sensitivity between datasets with different numbers of replicates (while controlling for the same number of total cells) is more pronounced.

Additionally, another potential reason for the observation made by the reviewer is that, in these particular simulations, for a very low number of cells per replicate (which is the case for datasets with a higher number of replicates), the potentially inaccurate inference of the parameters of the NB distribution lies upstream of DE detection and is systematic across all the replicates. In other words, it is possible that the introduced error in the estimates is biased in the same way across all the replicates. This potentially explains why DE detection in these cases cannot be simply rescued by a higher number of replicates.

We have extended Supplementary Note 2 to incorporate a slightly more in-depth analysis of the simulations, that includes the discussion above. Specifically, we added this paragraph:

As expected, sensitivity and FDR are highly dependent on the number of cells and replicates tested (Supplementary Note 2, Fig. 1A, B). Specifically, for all simulated datasets (i.e. varying numbers of control and case replicates), the average number of cells per replicate and total number of replicates appear to be the dominant factors, with sensitivity being positively scaled with the average number of cells per replicate and number of replicates (Supplementary Note 2, Fig. 1A, second and fourth columns). While controlling for the total number of cells, we observe that the number of replicates and class imbalance also affect DE detection. Specifically, higher class imbalance results in lower sensitivity, suggesting that the minimum number of cells (across the conditions) is the bottleneck for sensitivity in DE detection (Supplementary Note 2, Fig. 1B). Intriguingly, when the number of control and case replicates is the same, the total number of cells across all replicates is a key factor for DE detection (Supplementary Note 2, Fig. 1A, first column). Indeed, while controlling for the total number of cells, we observe that the difference between datasets with a variable number of replicates (relative to datasets with a balanced number of replicates across two conditions) is modest (Supplementary Note 2, Fig. 1A, inset). This can be explained by the absence of batch effects in the simulations, which leads to a higher marginal variance across replicates compared to the variance between cells within each replicate (this is particularly pronounced for pseudo-bulks consisting of fewer cells). In turn, when the variance within the replicates is much larger than the variance across the replicates, the low detection rate for small pseudo-bulks cannot be rescued by the higher number of replicates. Accordingly, when

introducing batch effects between replicates (thus increasing the variability), we observe a stronger impact of replicate number on DE detection (Supplementary Note 2, Fig. 1C). Finally, we note that if the detection of DE genes is defined solely by the p-value, the FDR plateaus around 0.15 and remains high across a wide range of total cell number (Supplementary Note 2, Fig. 1A, first and second columns). It has been previously shown that leveraging both the statistical significance and the effect size in DE analysis yields better gene ranking and stronger biological insights (McCarthy and Smyth 2009; Xiao et al. 2014). Accordingly, when we define DE detection based on both p-value and the expected minimum effect size, we observe a steady decrease in FDR with the higher total number of cells (Supplementary Note 2, Fig. 1A, third and fourth columns).

Finally, we also want to comment on the Wang 2019 reference. In this study, the authors benchmark several DE methods, either specifically designed for single-cell analysis or borrowed from analysis for bulk RNA-seq (including edgeR). To our initial understanding of the analysis performed in this paper, bulk methods were adapted as pseudo-bulks, with the aggregation of counts from cells across single replicates. However, after a careful re-reading of the article, we realise that edgeR was applied in a single-cell manner as well, and therefore, as pointed out by the reviewer, Wang 2019 does not support the statement we made in the manuscript. We apologise for this oversight. We still claim that the number of cells per replicate in pseudo-bulk methods has been previously shown to affect DE detection, and we now provide a different reference (Crowell et al. 2020) that supports the originally made statement.

Minor:

1. In Methods section 2.2, line 30 should read "no gene selection will [be] performed"
2. Is the "bulk DE" approach mentioned in the caption to Supplementary Figure 6, actually meant to say "pseudo-bulk DE"?

We have incorporated these changes in the revised submission.

References:

- Adams, Taylor S., Jonas C. Schupp, Sergio Poli, Ehab A. Ayaub, Nir Neumark, Farida Ahangari, Sarah G. Chu, et al. 2020. "Single-Cell RNA-Seq Reveals Ectopic and Aberrant Lung-Resident Cell Populations in Idiopathic Pulmonary Fibrosis." *Science Advances* 6 (28): eaba1983.
- Campbell, Kieran R., and Christopher Yau. 2018. "Uncovering Pseudotemporal Trajectories with Covariates from Single Cell and Bulk Expression Data." *Nature Communications* 9 (1): 2442.

- Crowell, Helena L., Charlotte Soneson, Pierre-Luc Germain, Daniela Calini, Ludovic Collin, Catarina Raposo, Dheeraj Malhotra, and Mark D. Robinson. 2020. "Muscat Detects Subpopulation-Specific State Transitions from Multi-Sample Multi-Condition Single-Cell Transcriptomics Data." *Nature Communications* 11 (1): 6077.
- Hou, Wenpin, Zhicheng Ji, Zeyu Chen, E. John Wherry, Stephanie C. Hicks, and Hongkai Ji. 2023. "A Statistical Framework for Differential Pseudotime Analysis with Multiple Single-Cell RNA-Seq Samples." *Nature Communications* 14 (1): 7286.
- McCarthy, Davis J., and Gordon K. Smyth. 2009. "Testing Significance Relative to a Fold-Change Threshold Is a TREAT." *Bioinformatics* 25 (6): 765–71.
- Morse, Christina, Tracy Tabib, John Sembrat, Kristina L. Buschur, Humberto Trejo Bittar, Eleanor Valenzi, Yale Jiang, et al. 2019. "Proliferating SPP1/MERTK-Expressing Macrophages in Idiopathic Pulmonary Fibrosis." *The European Respiratory Journal: Official Journal of the European Society for Clinical Respiratory Physiology* 54 (2). <https://doi.org/10.1183/13993003.02441-2018>.
- Reyfman, Paul A., James M. Walter, Nikita Joshi, Kishore R. Anekalla, Alexandra C. McQuattie-Pimentel, Stephen Chiu, Ramiro Fernandez, et al. 2019. "Single-Cell Transcriptomic Analysis of Human Lung Provides Insights into the Pathobiology of Pulmonary Fibrosis." *American Journal of Respiratory and Critical Care Medicine* 199 (12): 1517–36.
- Van den Berge, Koen, Hector Roux de Bézieux, Kelly Street, Wouter Saelens, Robrecht Cannoodt, Yvan Saeys, Sandrine Dudoit, and Lieven Clement. 2020. "Trajectory-Based Differential Expression Analysis for Single-Cell Sequencing Data." *Nature Communications* 11 (1): 1201.
- Xiao, Yufei, Tzu-Hung Hsiao, Uthra Suresh, Hung-I Harry Chen, Xiaowu Wu, Steven E. Wolf, and Yidong Chen. 2014. "A Novel Significance Score for Gene Selection and Ranking." *Bioinformatics* 30 (6): 801–7.

Second round of review

Reviewer 1

Looks great! All of my comments were addressed. Thanks!